# Apical contacts stemming from incomplete delamination guide progenitor cell allocation through a dragging mechanism

Eduardo Pulgar[1,2], Cornelia Schwayer[3], Néstor Guerrero[1,2], Loreto López[1,2], Susana Márquez[4], Steffen Härtel[1,2,5], Rodrigo Soto[4], Carl-Philipp Heisenberg[3], Miguel L Concha[1,2,6]*

[1]Institute of Biomedical Sciences, Faculty of Medicine, Universidad de Chile, Santiago, Chile; [2]Biomedical Neuroscience Institute, Santiago, Chile; [3]Institute of Science and Technology Austria, Klosterneuburg, Austria; [4]Physics Department, FCFM, Universidad de Chile, Santiago, Chile; [5]National Center for Health Information Systems, CENS, Santiago, Chile; [6]Center for Geroscience, Brain Health and Metabolism, Santiago, Chile

*For correspondence:
mconcha@uchile.cl

Competing interest: The authors declare that no competing interests exist.

**Abstract** The developmental strategies used by progenitor cells to allow a safe journey from their induction place towards the site of terminal differentiation are still poorly understood. Here, we uncovered a mechanism of progenitor cell allocation that stems from an incomplete process of epithelial delamination that allows progenitors to coordinate their movement with adjacent extra-embryonic tissues. Progenitors of the zebrafish laterality organ originate from the superficial epithelial enveloping layer by an apical constriction process of cell delamination. During this process, progenitors retain long-lasting apical contacts that enable the epithelial layer to pull a subset of progenitors on their way to the vegetal pole. The remaining delaminated cells follow the movement of apically attached progenitors by a protrusion-dependent cell-cell contact mechanism, avoiding sequestration by the adjacent endoderm, ensuring their collective fate and allocation at the site of differentiation. Thus, we reveal that incomplete delamination serves as a cellular platform for coordinated tissue movements during development.

## Introduction

During embryo development, naïve cell lineages undergo concurrent processes of fate specification and morphogenesis as critical steps towards the generation of differentiated tissues and organs. These early progenitor cells often travel long distances from their induction site to the site of terminal differentiation, making them vulnerable to environmental cues and movement of neighbouring tissues. When faced with these environmental hazards, progenitors may deviate from their correct pathway or change their potential, leading to a reduction in the pool of progenitors available for later stages of differentiation. The success of this journey is critical when the number of progenitors in a tissue or organ is limited. In these cases, small reductions in the number of progenitors can lead to developmental abnormalities resulting in a dysfunctional organ (*Moreno-Ayala et al., 2020*). Several embryonic tissues and organs originate from small sets of progenitor cells. These include the primordial germ cells that give rise to gametes in vertebrate and invertebrate gonads, the posterior lateral line primordium that give rise to neuromasts along the trunk and tail of fish and amphibians, and the progenitors of the laterality organ involved in left-right pattern formation in various vertebrates

(reviewed in *Dalle Nogare and Chitnis, 2017*; *Matsui and Bessho, 2012*; *Reig et al., 2014*; *Richardson and Lehmann, 2010*). Despite the importance of the developmental pathways followed by these small groups of progenitor cells and their impact on the physiology of the organism, we still know little about the set of developmental strategies that progenitor cells deploy in vivo to overcome the challenges imposed by the environment as they travel to the site of terminal differentiation. Here, we examine this question during the early stages of morphogenesis of the embryonic laterality organ, the first organ to form during vertebrate development, using the zebrafish as a model organism.

The zebrafish laterality organ or left-right organiser is a transient embryonic structure of epithelial nature known as Kupffer's vesicle that contains motile cilia necessary for left-right axis determination (*Cooper and D'Amico, 1996*; *Essner et al., 2005*; *Kramer-Zucker et al., 2005*). This organ-like epithelial structure originates from a small group of 20–30 progenitors, known as dorsal forerunner cells (DFCs), which arise at the dorsal margin of the late blastula by a process of cell ingression that converts the superficial epithelial cells of the extra-embryonic enveloping layer (EVL) into deep mesenchymal-type DFCs (*Oteiza et al., 2008*). After ingression, DFCs move as a cell collective from their place of origin at the equator of the embryo towards the vegetal pole, reaching the terminal location for organ differentiation at the posterior tip of the notochord (*Figure 1A*; *Figure 1—video 1*; *Cooper and D'Amico, 1996*; *Oteiza et al., 2008*). On their journey, DFCs are positioned ahead of the margin of the deep cell layer (DCL), where specification signals and massive internalisation movements transform the marginal epiblast into the mesendoderm (*Figure 1B*; *Pinheiro and Heisenberg, 2020*). Despite the proximity to the DCL margin and the fact that DFCs share critical determinants with the mesendoderm (*Alexander and Stainier, 1999*; *Warga and Kane, 2018*), the movement and fate of DFCs do not appear to be affected by specification signals or *mass* internalisation movements of the mesendoderm, remaining separate from this cellular domain during their vegetal movement. Notably, DFCs follow the same vegetal ward direction of movement as the overlying EVL during epiboly (*Bruce and Heisenberg, 2020*) and appear to be physically connected to this extra-embryonic epithelial tissue, as revealed by the presence of puncta enriched in the tight junction (TJ)-associated zonula occludens one protein (ZO-1) at the DFC-EVL interface (*Ablooglu et al., 2010*; *Oteiza et al., 2008*). This observation raises the question of whether the putative DFC-EVL connections play a role in the movement of DFCs to the site of organ differentiation. This hypothesis has not yet been tested experimentally.

Here, we addressed this hypothesis by combining in vivo imaging and biomechanical manipulation in zebrafish embryos. We found that DFCs arise from the EVL by an apical constriction process of cell delamination. During this process, DFCs retain long-term physical connections with the EVL and yolk syncytial layer (YSL) through TJ-enriched apical attachments, at the time both extra-embryonic tissues spread to the vegetal pole in the movement of epiboly. We demonstrated that extra-embryonic tissue spreading is transmitted to DFCs through the apical attachments to drag their vegetal movements. As DFCs detach from the extra-embryonic cellular domains after completing delamination or undergoing division, long polarised protrusions and E-cadherin (E-cad)-mediated adhesion integrate detached cells to the vegetal movement of attached DFCs, avoiding sequestration by the endoderm and ensuring the vegetal motion of all progenitors as a group. Thus, we unveil a drag-mediated guidance mechanism of progenitor cell morphogenesis that relies on the concerted activities of the epithelial delamination process underlying progenitor cell specification and the directed spreading of adjacent extra-embryonic tissues.

## Results

### DFCs ingress by cell delamination through an apical constriction process that provides long-term apical attachments with extra-embryonic tissues

The previous observation of puncta enriched in the TJ adaptor protein ZO-1 at the interface between a subset of DFCs and the overlying epithelial EVL during late epiboly stages (*Ablooglu et al., 2010*; *Oteiza et al., 2008*) suggests that DFCs establish contacts that attach them to the extra-embryonic surface epithelium. To investigate this possibility, we started by studying the spatial distribution of these enriched ZO-1 regions. Double immunolabelling for ZO-1 and phalloidin at 75 % epiboly confirmed that ZO-1 puncta marked discrete areas, also enriched in F-actin, where DFCs contacted the EVL at

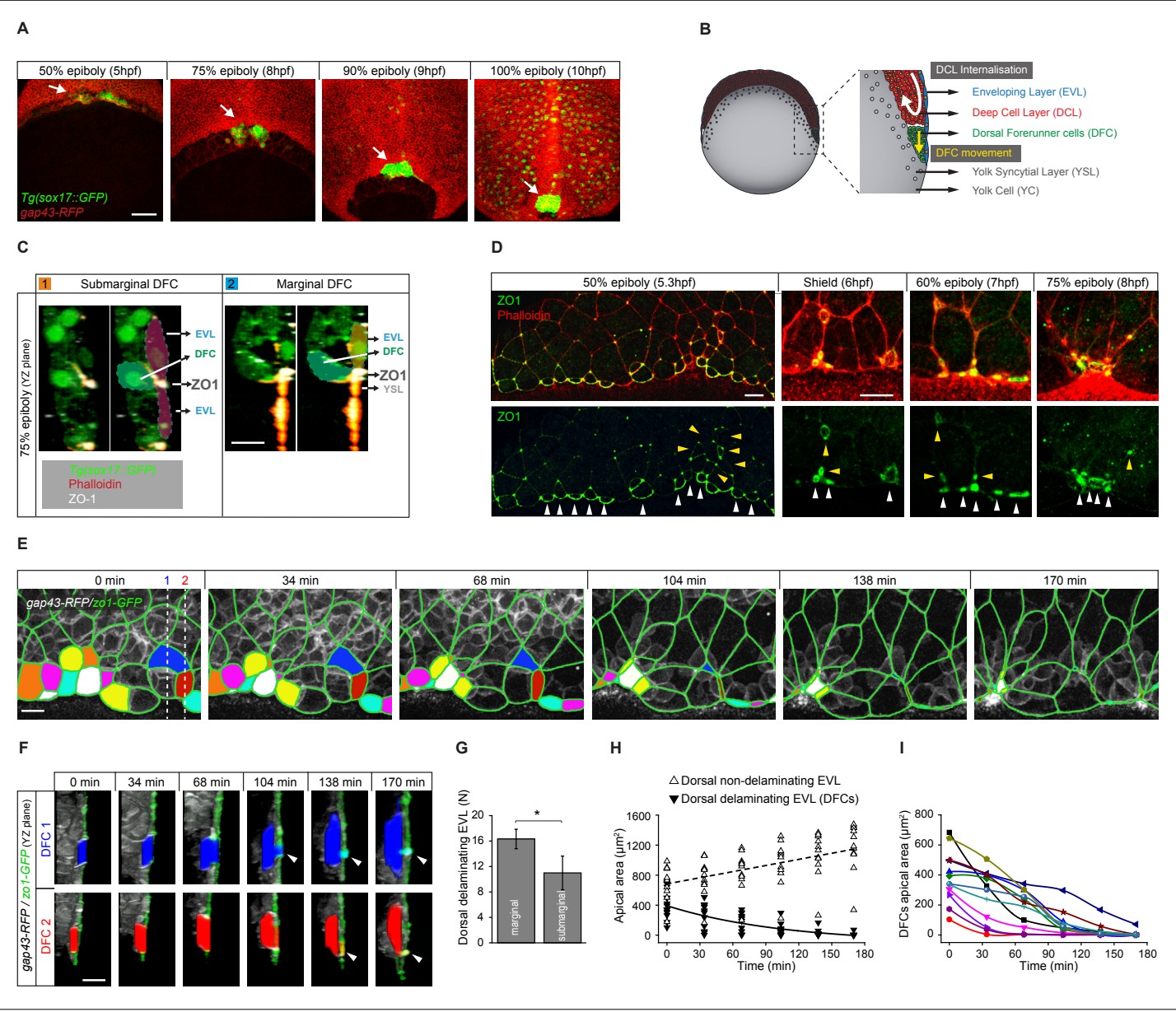

**Figure 1.** Dorsal forerunner cells (DFCs) delaminate by apical constriction and retain apical attachments with the enveloping layer (EVL) and yolk syncytial layer (YSL). (**A**) Dorsal views of confocal z-stack maximum projections showing the collective vegetal movement of DFCs between shield stage and 100 % epiboly in a representative Tg(*sox17::GFP*) embryo injected with *gap43-RFP* mRNA. DFCs are in green (arrows) while the plasma membrane of all cells is in red. Note that the *sox17::GFP* transgene also labels the scattered population of endodermal cells at advanced stages of epiboly (see criteria to discriminate DFCs from endodermal cells in Material and methods) (extracted from *Figure 1—video 1*). Scale bar, 100 µm. (**B**) Schematic diagram of a cross section along the sagittal plane of the zebrafish embryo at 60 % of epiboly. DFCs move to the vegetal pole ahead of the deep cell layer (DCL) margin, where mesendodermal progenitors internalise. (**C**) Confocal microscopy zy-plane of a 75 % epiboly Tg(*sox17::GFP*) embryo (green) stained with phalloidin (red) and zonula occludens one protein (ZO-1) (white), showing submarginal (left) and marginal (right) DFCs connected with the EVL and YSL by focal apical attachments enriched in ZO-1 and F-actin (extracted from *Figure 1—figure supplement 1*). Scale bar, 20 µm. (**D**) Phalloidin and ZO-1 immunostaining (merge on top and ZO-1 on bottom) of the dorsal margin of wild-type embryos between 50% and 75% epiboly. Images correspond to surface confocal sections showing the apical domains of marginal (white arrowheads) and submarginal (yellow arrowheads) delaminating DFCs in contact with the YSL and EVL. Scale bar, 20 µm. (**E**) Time series of dorsal views of confocal z-stack maximum projections of a representative embryo injected with *zo1-GFP* and *gap43-RFP* between 50% and 80% epiboly, showing EVL cell junctions (green outlines) and the apical domains of EVL cells as they delaminate to become DFCs (coloured areas) (extracted from *Figure 1—video 2*). Scale bar, 20 µm. (**F**) Time series of confocal z-sections showing two delaminating EVL cells (DFCs) taken from panel E in lateral views (blue and red cells) as they move below the plane of the EVL epithelium during the process of delamination. Note that delaminating cells retain a focal apical attachment with the EVL (arrowhead, top) and YSL (arrowhead, bottom). Scale bar, 20 µm. (**G**) Quantification of the number of dorsal EVL cells undergoing delamination to become DFCs at both marginal

*Figure 1 continued on next page*

*Figure 1 continued*

and submarginal positions, expressed as means ± s.d. *p ≤ 0.05 (n = 49 marginal DFCs and 33 submarginal DFCs from three embryos). (**H**) Temporal changes in apical area of delaminating dorsal EVL cells (DFCs) (black triangles) and neighbouring non-delaminating dorsal EVL cells (white triangles) in a representative embryo during the process of delamination (n = 12 DFCs and 11 dorsal EVL cells from the representative embryo shown in E). Continuous and dashed lines indicate the mean values of apical area of delaminating and non-delaminating dorsal EVL cells, respectively. (**I**) Temporal changes in apical area of individual delaminating DFCs in a representative embryo during the process of delamination (n = 12 delaminating DFCs from the representative embryo shown in E). Each curve corresponds to a single cell. Animal is to the top in all image panels. Source data for all plots are provided in *Figure 1—source data 1*.

The online version of this article includes the following video, source data, and figure supplement(s) for figure 1:

**Source data 1.** Source data for *Figure 1*.

**Figure 1—video 1.** Movement of dorsal forerunner cells (DFCs) from the embryo equator to the vegetal pole during epiboly (related to Figure 1A).
https://elifesciences.org/articles/66483/figures#fig1video1

**Figure 1—video 2.** Delamination, apical constriction, and vegetal movement of dorsal forerunner cells (DFCs) during epiboly (related to Figure 1E and F).
https://elifesciences.org/articles/66483/figures#fig1video2

**Figure supplement 1.** Zonula occludens one protein (ZO-1) and F-actin accumulate at dorsal forerunner cell (DFC)-enveloping layer (EVL) cell junctions.

cell-cell junctions near the margin of the epithelium (*Figure 1C* – left; *Figure 1—figure supplement 1*). Also, we observed the punctuated co-accumulation of ZO-1 and F-actin at the epithelial margin where DFCs contacted the YSL (*Figure 1C* – right; *Figure 1—figure supplement 1*). Indeed, the latter population of marginal DFCs was more abundant than submarginal DFCs (*Figure 1D and G*). We then asked how these contacts arise by examining the temporal progression of ZO-1 from early to late epiboly stages throughout DFC formation. Previous work has shown that DFCs originate from dorsal EVL cells through ingression (*Oteiza et al., 2008*) although the underlying mechanism is currently unknown. We found that at early epiboly stages EVL cells fated to become DFCs accumulated ZO-1 along their apical junctions (*Figure 1D*). Later, as epiboly progressed, the apical face of these cells gradually reduced in area to leave punctuated apical domains enriched in ZO-1 and F-actin proteins (*Figure 1D*). Together, these findings reveal that DFCs arise from dorsal EVL cells by a mechanism of epithelial cell delamination mediated by apical constriction. Remarkably, this process leaves discrete apical attachments connecting delaminating DFC progenitors with the extra-embryonic tissues.

To investigate the dynamics of apical constriction leading to the formation of apical attachments, we performed in vivo imaging from late blastula to epiboly stages using a GFP-tagged version of ZO-1. We found that EVL cells fated to become DFCs gradually shrunk their apical area while moving beneath the epithelial sheet (*Figure 1E and F*; *Figure 1—video 2*), a behaviour that was specific as neighbouring dorsal EVL cells exhibited the opposite behaviour, increasing their apical area as epiboly progressed (*Figure 1H*). Interestingly, the kinetics of apical area reduction varied among cells (*Figure 1I*) and resulted in discrete apical attachments that connected these cells with the YSL and EVL for extended periods (*Figure 1F*). Collectively, these findings indicate that DFC progenitors retain long-term apical attachments with the extra-embryonic YSL and EVL as a consequence of the apical constriction process of cell delamination that gives rise to the cells.

## Cell delamination is concurrent with the vegetal movement of DFCs and extra-embryonic tissues

The presence of apical attachments connecting delaminating DFCs with the YSL and EVL opens the possibility that extra-embryonic tissues guide the vegetal movement of these cells. However, previous studies have hypothesised that the vegetal movement of DFCs relies on polarised protrusive activity (*Ablooglu et al., 2010*; *Zhang et al., 2016*). To discriminate between these two guidance mechanisms, we first examined the organisation of polarised protrusions in DFCs. Using in vivo imaging in an actin transgenic reporter line, we found that DFCs formed dynamic membrane protrusions (*Figure 2E*), as previously observed (*Ablooglu et al., 2010*; *Zhang et al., 2016*). However, protrusions were enriched at the rear (lateral and animal) edges and not at the leading (vegetal) edge of the DFC cluster (*Figure 2A–D*) arguing against a primary role of polarised cell protrusions in directing the vegetal movement of DFCs. To further support this idea, we inhibited the formation of actin-based cell protrusions through the expression of a dominant negative form of Rac (Rac1-T17N) (*Bakkers*

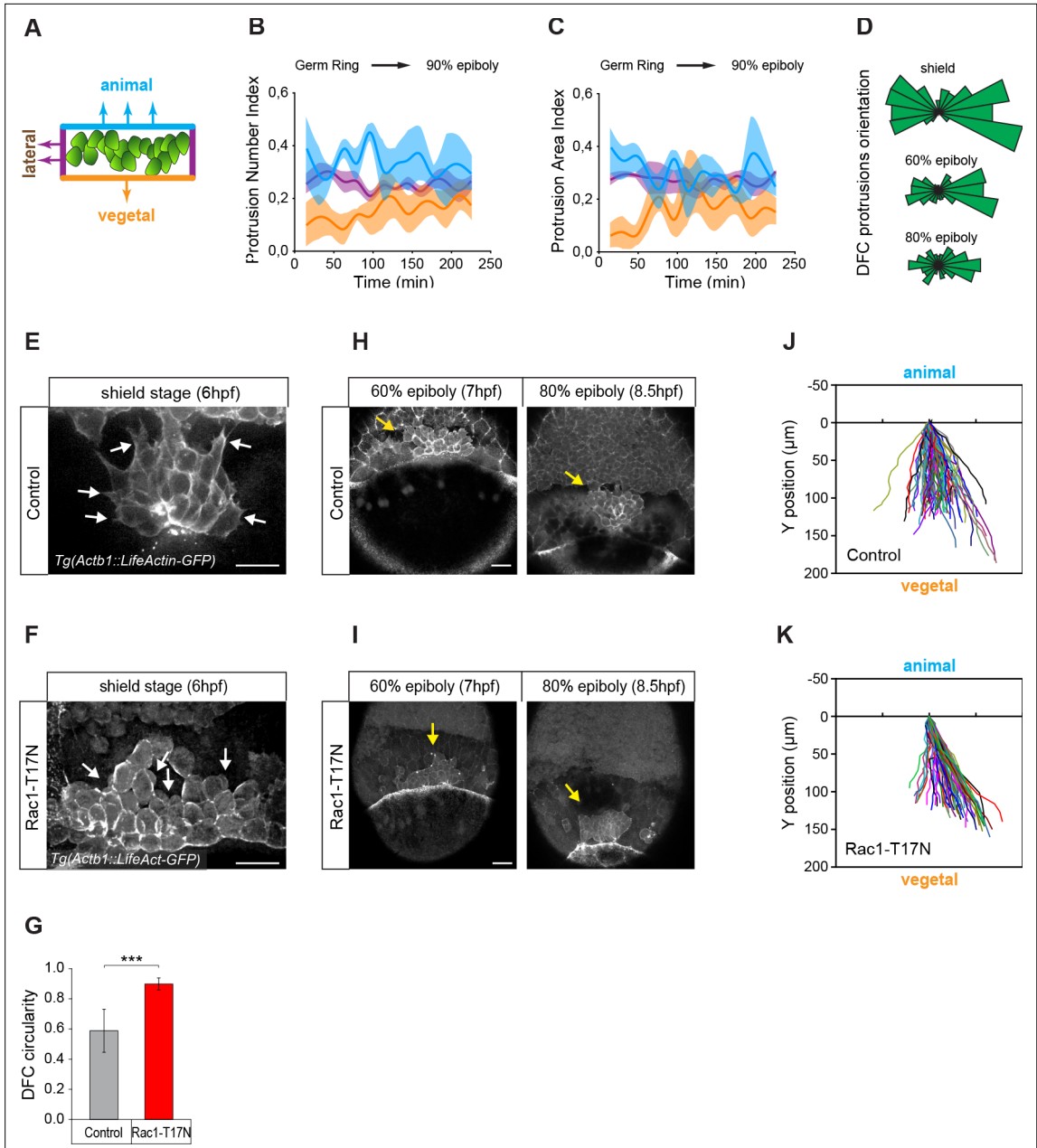

**Figure 2.** Vegetally directed polarised protrusions are not required for dorsal forerunner cell (DFC) vegetal movement. (**A**) Schematic representation of the DFC cluster showing how cell protrusions extending from the vegetal (orange), lateral (purple), and animal (light blue) edges of the cluster were quantified in Tg(*actb1::lifeactin-RFP*) embryos to build the plots of (**B**) and (**C**). (**B**) Kinetic of normalised DFC protrusion number in living embryos (n = 3 embryos). (**C**) Kinetic of normalised DFC protrusion area in living embryos (n = 3 embryos). (**D**) Circular distribution plots of DFC protrusion orientation at different developmental stages obtained from fixed Tg(*sox17::utrn-GFP*) embryos (n = 11, 10, and 9 embryos for shield, 60 % epiboly and 80 % epiboly, respectively). (**E, F**) Dorsal views of confocal z-stack maximum projections showing the protrusions formed in DFCs (white arrows) from representative living Tg(*actb1::lifeactin-RFP*) embryos at shield stage in control (**E**) and Rac1-T17N injected (**F**) conditions. Scale bar, 20 μm. (**G**) Plot of circularity index of DFC protrusions in control and Rac1-T17N injected embryos, expressed as means ± s.d. ***p ≤ 0.001 (n = 42 cells from two control embryos and 62 cells from 3 Rac1-T17N embryos). (**H, I**) Dorsal views of confocal z-stack maximum projections showing the vegetal movement of the DFC cluster (yellow arrows) from representative living Tg(*actb1:lifeactin-RFP*) embryos from 60% to 80% epiboly stages in control (**H**) and Rac1-T17N injected (**I**) conditions. Scale bar, 20 μm. (**J, K**) Tracking plots of DFC movement in control (**J**) and Rac1-T17N injected (**K**) conditions, showing the directional movement of DFCs (n = 2 embryos for control and 3 embryos for Rac1-T17N injected conditions). Animal is to the top in all image panels. Source data for all plots are provided in *Figure 2—source data 1*.

The online version of this article includes the following source data for figure 2:

**Source data 1.** Source data for *Figure 2*.

*et al., 2004*). DFCs expressing Rac1-T17N did not form lamellar nor filopodial-like protrusions as in controls but instead showed extensive membrane blebbing with no directional bias (*Figure 2F*). Nevertheless, they still retained a persistent collective movement towards the vegetal pole (*Figure 2I and K*) as in controls (*Figure 2H and J*).

Since the vegetal movement of DFCs does not seem to rely on a guidance mechanism based on vegetally directed polarised cell protrusions, we tested whether apical attachments could play a key role in this process. For this, we first determined the extent to which the vegetal movement of DFCs temporally matched the process of delamination using confocal microscopy (*Figure 3A*). We observed that DFCs travelled over 350 µm towards the vegetal pole from the time in which the first cell started the process of apical constriction at around dome stage until the last cells delaminated after detachment from the extra-embryonic tissues at around 90 % epiboly (*Figure 3B* – horizontal axes). In agreement with the variable kinetics of apical constriction (*Figure 1I*), the onset of apical constriction and the time of detachment leading to delamination were also highly variable in the DFC population (*Figure 3B and C* – red and blue circles, respectively). Together, the absence of vegetal ward polarised cell protrusions in DFCs and the close concurrency of cell delamination with the vegetal movement of DFCs suggest that apical attachments with the YSL and EVL integrate delaminating DFCs to the vegetalward movements of the extra-embryonic tissues during epiboly. In support of this idea, the simultaneous tracking of DFCs and EVL cells revealed that the speed, direction, and extent of DFC vegetal movement tightly matched the vegetal spreading of the EVL (*Figure 3D–G*; *Figure 3—video 1*). Furthermore, DFCs and overlying EVL cells shared the same bias in the orientation of the elongation axis (*Figure 3—figure supplement 1*), indicating a morphogenetic coupling between the two tissues. Collectively, these results suggest that apical attachments derived from the process of cell delamination work as tissue-tissue connectors that couple delaminating DFCs with the vegetal spreading of the extra-embryonic YSL and EVL, guiding their vegetal movement during epiboly.

## Extra-embryonic tissues pull DFCs through apical attachments to guide their vegetal movement

To directly test if apical attachments between DFCs and the extra-embryonic tissues guide the vegetal movement of DFCs, we first disrupted the vegetal spreading of the YSL/EVL. Previous reports have shown that contraction and friction-based flows of a ring-like YSL actomyosin network drive YSL/EVL vegetal spreading during epiboly (*Behrndt et al., 2012*; *Cheng et al., 2004*; *Heisenberg and Bellaiche, 2013*). Therefore, we inhibited the activity of the YSL actomyosin ring by decreasing the levels of phosphorylated myosin II through expressing, specifically in the YSL, the N-terminal region of the myosin phosphatase target subunit 1 (N-ter-MYPT1) (*Figure 4—figure supplement 1A,B*). Embryos expressing N-ter-MYPT1 (MYPT1) showed a speed drop in YSL/EVL vegetalward spreading. Likewise, DFCs decreased their vegetal ward speed to the same extent (*Figure 4—figure supplement 1C,D*). Similarly, local disruption by laser ablation of the YSL actomyosin ring in direct contact with marginal DFC apical attachments decreased the speed of EVL spreading, as previously shown (*Behrndt et al., 2012*), simultaneously decreasing the speed of DFC vegetal movement near the ablated zone (*Figure 4A and B*; *Figure 4—video 1*). Together, these results demonstrate that the progress of DFC vegetal movement requires the vegetal spreading of the extra-embryonic YSL/EVL.

Next, we conducted two types of experiments to assess if extra-embryonic tissue movement is transmitted to DFCs through their apical attachments. As a first experiment, we transiently released a subset of DFC apical attachments from the influence of EVL spreading by performing laser ablation of a cortical EVL junction located immediately vegetal to DFC apical attachments (*Figure 4C*). We observed that apical attachments recoiled towards the animal pole after ablation, triggering a fast retraction of the DFC body, indicating that both apical attachments and DFCs were under vegetally directed tension (*Figure 4D and E*; *Figure 4—video 2*). Subsequently, the constricted cortical zone induced by wound repair pulled apical attachments towards the vegetal pole, restoring the vegetal directionality of DFC movement, further indicating that apical attachments are able to transmit motion information from the EVL to DFCs (*Figure 4D and E*; *Figure 4—video 2*). Finally, we took advantage of the particular spatial configuration that DFCs adopt in some wild-type embryos, in which subsets of cells developed as individuals isolated from the core group of DFCs, to test the requirement of apical attachments for DFC directed vegetal movements (*Figure 5A*). In these 'natural' experiments, we observed that isolated individual DFCs devoid of apical attachments (hereafter referred to as

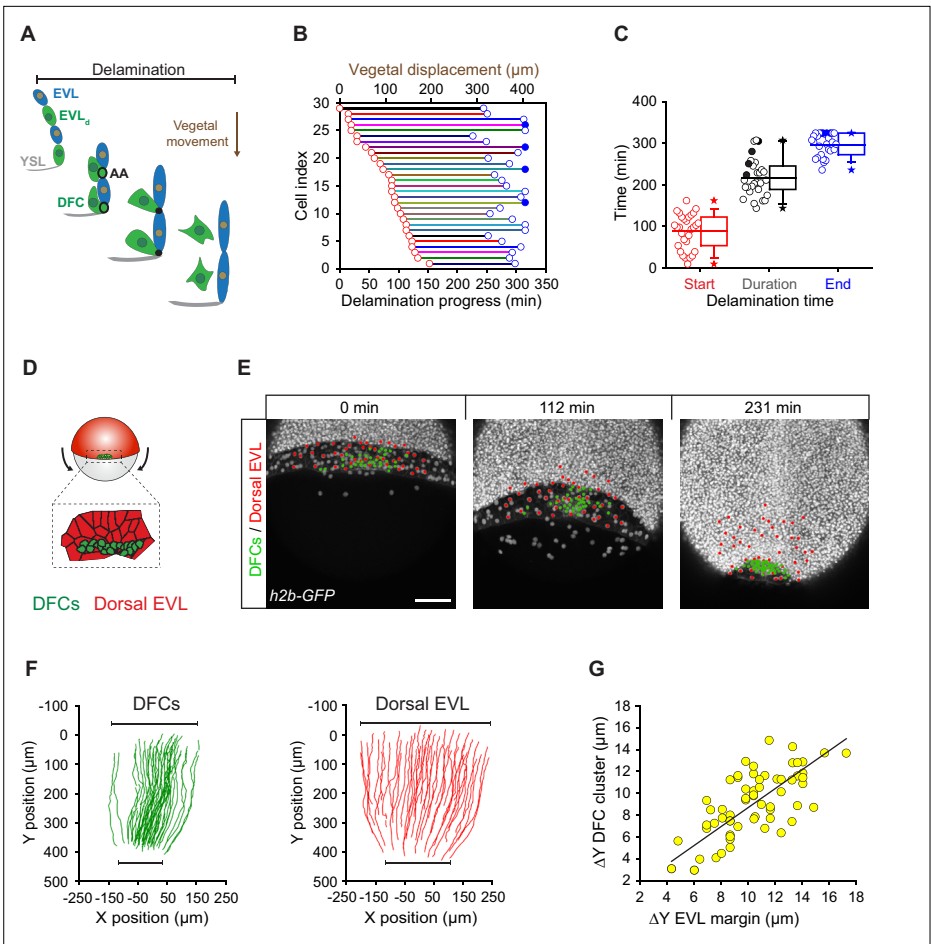

**Figure 3.** Dorsal forerunner cell (DFC) delamination is asynchronous and coexists with the vegetal movement of DFCs and the enveloping layer (EVL). (**A**) Schematic diagram showing the origin of DFCs from the EVL through cell delamination. Apical attachments (AA) that result from apical constriction connect delaminating DFCs with the EVL and YSL during their vegetal movements. When DFCs complete delamination, they lose apical attachments and are released from the EVL and YSL. (**B**) Plot depicting the relationship between the time of the DFC delamination process and the vegetal movements of delaminating cells for all cells in a single representative Tg(*actb1::lifeactin-RFP*) embryo. The start (red circles) and end (blue circles) times of cell delamination, and the total duration of this process (horizontal lines) are shown for individual delaminating cells (bottom axis) and compared with their vegetal movements (top axis). Filled blue circles indicate a subset of delaminating cells that still retained their apical attachments by the end of the movies (n = 29 delaminating cells from one representative embryo). (**C**) Combined box and distribution plots of the start and end times, and of the total duration of cell delamination for the same representative embryo as in B. Circles correspond to individual values while the box depicts the interquartile range from 25% to 75% of the data around the average (vertical line inside the box), the whisker depicts s.d., and stars indicate maximum and minimum values. Filled blue and black circles indicate the subsets of delaminating cells that still retained apical attachments by the end of the movies (n = 29 delaminating cells from one representative embryo, same as in B). (**D–G**) Delaminating DFC mirrors the movement of dorsal EVL cells. Schematic diagram (**D**) and dorsal views of confocal z-stack maximum projections (**E**) of a representative wild-type embryo injected with *h2b-GFP* mRNA to label all nuclei (white dots) (extracted from *Figure 3—video 1*). (**F**) Tracks of delaminating cells (DFCs, green) and neighbouring non-delaminating dorsal EVL cells (red) between shield stage and 90 % of epiboly, showing the paths of vegetal movement and convergence. (**G**) Correlation plot between the changes in position along the y-axis of the centre of mass of the DFC cluster and the EVL margin (Pearson's r = 0.97552) (n = 62 tracked delta time points from three embryos). Scale bar, 100 μm. Animal is to the top in all image panels. Source data for all plots are provided *Figure 3—source data 1*.

The online version of this article includes the following video, source data, and figure supplement(s) for figure 3:

**Source data 1.** Source data for *Figure 3*.

**Figure 3—video 1.** The movement of dorsal forerunner cells (DFCs) parallels the movement of the enveloping cell

*Figure 3 continued on next page*

*Figure 3 continued*
layer in zebrafish (related to Figure 3D–G).
https://elifesciences.org/articles/66483/figures#fig3video1

**Figure supplement 1.** Alignment of dorsal forerunner cell (DFC) first principal axis mirrors the alignment of first principal axis of neighbouring enveloping layer (EVL) cells.

'detached DFCs') moved without persistence and directionality, even towards the animal pole in the opposite direction to the vegetal movement of the YSL/EVL (*Figure 5B–E*; *Figure 4—video 3*). In contrast, isolated individual DFCs connected with the extra-embryonic tissues by apical attachments (hereafter referred to as 'attached DFCs') showed a persistent and directed movement toward the vegetal pole, mimicking the movement of extra-embryonic tissues (*Figure 5B–E*; *Figure 4—video 3*). Importantly, the persistence and speed of movement of DFCs correlated with the persistence and speed of movement of apical attachments (*Figure 5G and H*), and this close spatial and temporal relationship between apical attachments and DFC movements (*Figure 5F–I*) was only lost upon detachment (*Figure 5E*; *Figure 4—video 3*), indicating the requirement of physical attachment to extra-embryonic tissues for the persistent directed vegetal movement of DFCs. Taken together, these findings demonstrate that apical attachments are required to transmit vegetal spreading from extra-embryonic tissues to attached DFCs, guiding their vegetal movement during epiboly.

## DFCs sustain a collective vegetal movement despite the increase of detached cells

The tracking of single isolated detached DFCs revealed that they failed to engage in directed and persistent vegetal movements (*Figure 5A–E and J*; *Figure 4—video 3*). Importantly, this behaviour was also present in a subset of detached cells initially located close to or inside the central cluster of DFCs during epiboly. We observed that in 55 % of embryos, often one or two, and up to five detached DFCs per embryo left the cluster as single cells (*Figure 6A*). These cells moved mainly in direction to the animal pole, being sequestered by the internalisation movement of the DCL and incorporating into this embryonic cell layer (*Figure 6B*; *Figure 6—video 1*). Remarkably, this group of DFCs mimicked the morphology and movement behaviour of endodermal cells and at later stages differentiated in endodermal tissue derivatives (*Figure 6—figure supplement 1A*). Furthermore, DFCs transplanted into the paraxial region of host embryos, where endodermal and mesodermal progenitors develop, also followed endodermal cell fate and form endodermal tissue derivatives (*Figure 6—figure supplement 1B*). Thus, detached DFCs are prone to leave the cell cluster being sequestered by the DCL, and in this new environment integrate into the endodermal cell layer, losing their primal fate. However, this behaviour was barely seen in normal development. Indeed, detached DFCs following the endodermal path were observed in only 55 % of all analysed embryos and represented a small fraction of the complete pool of detached DFCs (*Figure 6A*). In contrast, the number of detached DFCs increased steadily during epiboly as cells completed the process of delamination (*Figure 3B*). These findings prompted us to investigate how the detached DFC population integrates into the collective movement of attached DFCs, which is guided by the extra-embryonic tissues. To address this issue, we first analysed how the detached DFC population evolves in number during their collective vegetal movement. Immunostaining for ZO-1 and F-actin revealed that at the onset of epiboly the entire population of DFCs were transiting the process of delamination and thus were all attached to the YSL and EVL (*Figure 6C*). However, as epiboly progressed, detached DFCs steadily increased to outnumber the attached DFC population from around 75 % epiboly, and by 90 % epiboly they became the predominant pool of DFCs (*Figure 6C*). Through time-lapse microscopy, we found that the progressive increase of detached cells was not only due to DFCs completing the process of delamination (*Figure 3B*), but detached DFCs also emerged from events of cell division (*Figure 6D and E*). As previously noted, despite the continuous increase in the number of detached DFCs (*Figure 6E*), the entire population of DFCs still moved in direction to the vegetal pole in coordination with the vegetal spreading of extra-embryonic tissues until advanced stages of epiboly (*Figure 6F*). Only around 85–90% of epiboly, when the percentage of detached cells reached 80%, the DFC cluster detached from the EVL and significantly lowered its vegetal movement (*Figure 6E and F*). Taken together, these results reveal that the DFC cluster retains a collective vegetal motion during a prolonged period of epiboly, regardless

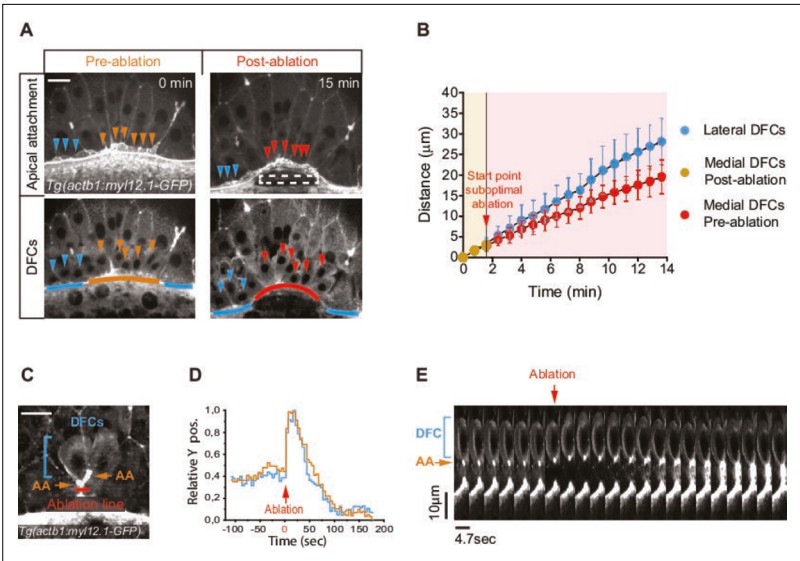

**Figure 4.** Apical attachments transmit extra-embryonic tissue spreading to guide dorsal forerunner cell (DFC) vegetal movement. (**A, B**) Laser disruption of the yolk syncytial layer (YSL) actomyosin network impairs DFC vegetal movements. (**A**) Dorsal views of a Tg(*actb1::myl12.1-GFP*) embryo at early shield stage (5.8 hpf) before and after laser ablation of the cortical actomyosin ring of the YSL (extracted from *Figure 4—video 1*). Confocal optical planes are at the level of the enveloping layer (EVL) to reveal the apical attachments (top) and at a deeper z-position to show the body (bottom) of delaminating DFCs. The dashed rectangle depicts the zone of laser ablation. Arrowheads point to apical attachments (top) and the nuclei of individual DFCs (bottom), while lines depict the position of the EVL margin immediately above the ablation zone (orange and red) and lateral to this region (blue). Scale bar, 20 µm. (**B**) Plot showing the changes in position of delaminating DFCs along the y-axis before and after ablation, with the pulses of laser ablation starting just before minute 2 (red arrow). Medial and lateral DFCs correspond to the cells shown in panel A (coloured arrowheads), whose apical attachments associate with the EVL/YSL margin above or lateral to the ablated YSL zone, respectively. Values correspond to means ± s.d. (**C–E**) Laser line ablation of an EVL cortical junction below the apical attachments of two submarginal delaminating DFCs in a Tg(*actb1::myl12.1-GFP*) embryo at 70 % of epiboly. (**C**) Dorsal view of a pre-ablation stage showing delaminating DFCs (blue bracket), their apical attachments (AA, orange arrows), and the ablation line (red). Scale bar, 20 µm. (**D**) Plot showing the changes in relative position of delaminating DFCs (blue) and apical attachments (orange) along the y-axis before and after ablation, with zero corresponding to the time of the laser pulse (red arrow). (**E**) Kymograph showing the movement of a DFC and its apical attachment during the laser ablation plotted in D (extracted from *Figure 4—video 2*). Scale bar, 10 µm. Animal is to the top in all image panels. Source data for all plots are provided in *Figure 4—source data 1*.

The online version of this article includes the following video, source data, and figure supplement(s) for figure 4:

**Source data 1.** Source data for *Figure 4*.

**Figure 4—video 1.** The progress of dorsal forerunner cell (DFC) vegetal movement requires the vegetal spreading of the extra-embryonic yolk syncytial layer/enveloping layer (YSL/EVL) (related to Figure 4A and B).
https://elifesciences.org/articles/66483/figures#fig4video1

**Figure 4—video 2.** Apical attachments of dorsal forerunner cells (DFCs) are under pulling tension from extra-embryonic tissues (related to Figure 4C–E).
https://elifesciences.org/articles/66483/figures#fig4video2

**Figure 4—video 3.** Apical attachments promote a persistent vegetal movement of attached dorsal forerunner cells (DFCs) (related to Figure 4).
https://elifesciences.org/articles/66483/figures#fig4video3

**Figure supplement 1.** Genetic disruption of the yolk syncytial layer (YSL) actomyosin network impairs dorsal forerunner cell (DFC) vegetal movement.

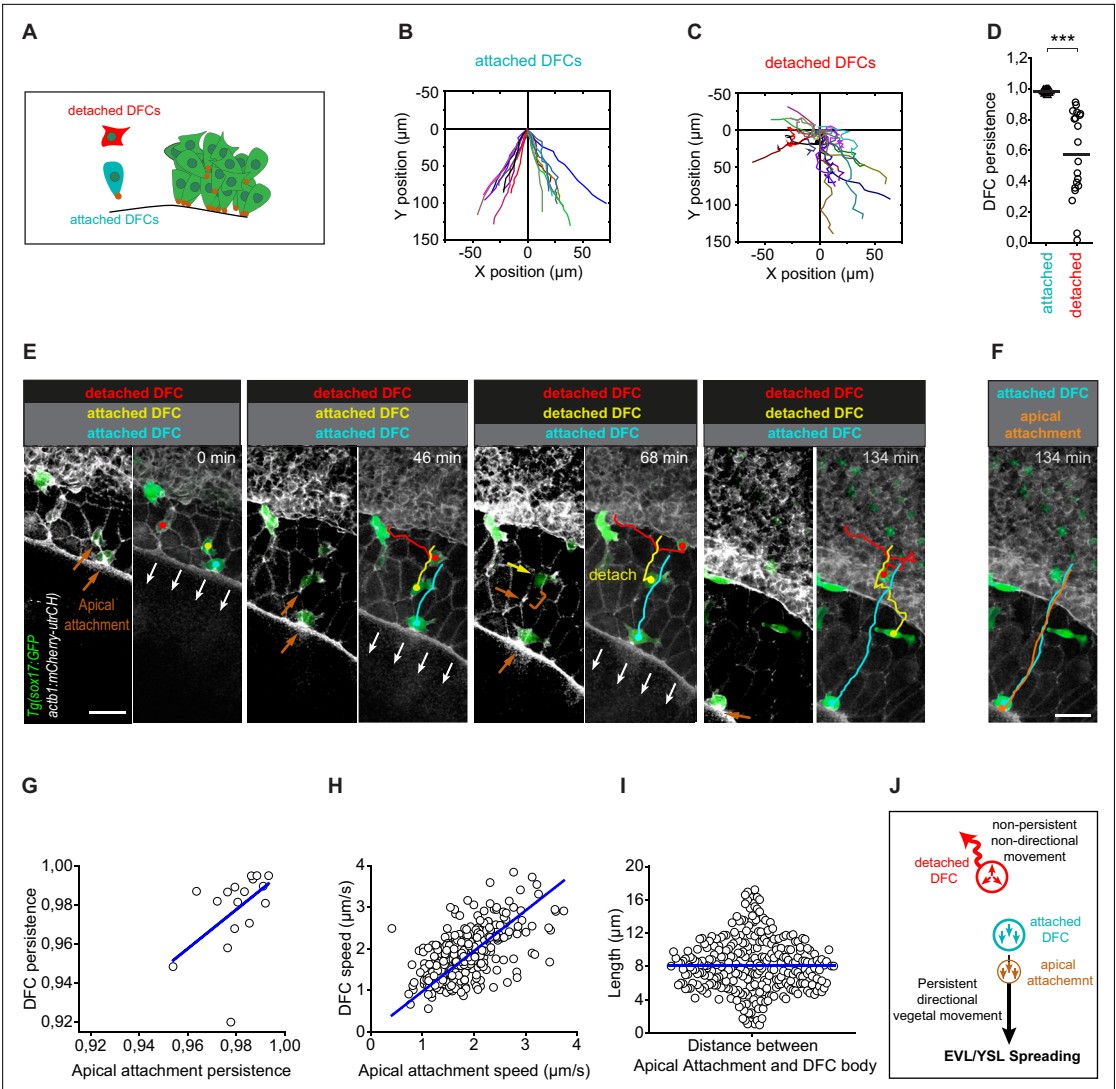

**Figure 5.** Apical attachments promote a persistent vegetal movement of attached dorsal forerunner cells (DFCs). (**A**) Schematic diagram showing single isolated DFCs transiting the process of delamination far from the main DFC cluster (green). Single isolated DFCs are either transiting delamination and be attached to extra-embryonic tissues (blue cell; attachment shown as a brown dot) or be detached from these tissues after completing delamination (red cell). (**B–D**) Movement directionality and persistence of single isolated attached and detached DFCs. Tracking plots of single isolated attached (**B**) and detached (**C**) DFCs, and the differences in movement persistence (**D**), obtained from living Tg(*sox17::GFP;actb1::mCherry-utrCH*) embryos at around 60 % epiboly (n = 17 attached DFCs from eight embryos and 20 detached DFCs from eight embryos). *** (p ≤ 0.001). (**E**) Time series of dorsal views of confocal z-stack maximum projections of a representative living Tg(*sox17::GFP;actb1::mCherry-utrCH*) embryo starting at 60 % of epiboly (extracted from *Figure 4—video 3*) showing the movement of three DFCs developing as single cells in isolation from the main DFC cluster (see schematic in panel A). The blue cell is attached to the extra-embryonic yolk syncytial layer (YSL) and enveloping layer (EVL) during the entire series. The red cell is detached from these tissues during the entire series. The yellow cell is initially attached to the EVL but at time 68 min completes delamination and detaches from the EVL. Note that after detaching, DFC movement loses persistence and becomes non-directional. Brown arrows show the accumulation of actin at apical attachments. The brown bracket at time 68 min shows the distance between the position of the apical actin accumulation that labelled the recently lost apical attachment and the position of the detached DFC. White arrows show the vegetal movement of the YSL/EVL margin during epiboly. (**F**) Comparison of the tracks of the movement of the blue attached DFC shown in (**E**) and the movement of its apical attachment (brown). Scale bar, 50 μm. (**G–I**) Correlation plots between the movement persistence of DFCs and their apical attachments (**G**; Pearson's r = 0.99984; n = 17 attached cells and apical attachments from eight embryos) and between the movement speed of DFCs and their apical attachments (**H**; Pearson's r = 0.96085; n = 276 attached cells and apical attachments from eight embryos), and distribution plot of the distance (length) between the apical attachment and body of DFCs (**I**; n = 276 apical attachments and DFC bodies from eight embryos). (**J**) Schematic diagram showing the proposed drag-mediated mechanism guiding the motion of attached DFCs. YSL/EVL vegetal spreading is transmitted to delaminating DFCs through apical attachments to guide their vegetal movements. Single isolated DFCs devoid of apical attachments (detached DFCs) are insensible to YSL/EVL dragging and show non-directional movements. Animal is to the top in all image panels. Source data for all plots are provided in *Figure 5—source data 1*.

*Figure 5 continued on next page*

*Figure 5 continued*

The online version of this article includes the following source data for figure 5:

**Source data 1.** Source data for *Figure 5*.

of the progressive expansion of the detached DFC population, raising the question of how detached DFCs move towards the vegetal pole.

## DFC-DFC contact interactions integrate detached cells to the vegetal movement of attached DFCs providing a clustered collective movement

The observation that most detached DFCs move towards the vegetal pole despite their ability to migrate out of the DFC cluster and be sequestered by the DCL suggests the existence of specific mechanisms that integrate these detached cells into the movement of attached DFCs. During epiboly, DFCs change from being a collection of dispersed progenitors to a tightly packed cellular cluster (*Figure 1A*; *Figure 1—video 1*; *Oteiza et al., 2008*). Quantitative assessment of collective cell behaviour revealed that DFC clustering was achieved by progressively decreasing the distance between progenitors and increasing the contact with the surrounding cells (*Figure 7A and B*). During this process, DFCs express *e-cadherin* mRNA (*Kane et al., 2005*) and also localise E-cad at the cell membrane (*Figure 7E*), and previous reports have shown that functional abrogation of this adhesion molecule affects the cluster cohesion of DFCs leading to a scattered organisation of DFCs at 90 % of epiboly (*Oteiza et al., 2010*). Thus, E-cad-mediated DFC-DFC adhesion could play a role of integrating detached DFCs into the vegetal movement of attached DFCs. Therefore, we tested by time-lapse microscopy whether the directed vegetal movement of detached DFCs requires E-cad-mediated contact interactions. We found that E-cad knockdown disrupted cluster cohesion, leading to multiple racemes of DFCs that moved in the vegetal direction guided by their apical attachments with the YSL and EVL (*Figure 7F*). Remarkably, the number of detached cells exiting the DFC group towards the DCL increased significantly in these embryos (*Figure 7G and H*). These results indicate that E-cad-mediated DFC-DFC adhesion is necessary for the recruitment of detached DFCs into the vegetal motion of attached DFCs, avoiding cell escape and promoting the formation of a tight cellular cluster.

We observed that the DFC population organised as a single cell cluster from early stages of vegetal movement in 57 % of embryos (*Figure 8A and C*). However, in the remaining 43 % of embryos, DFCs initially formed two (28%) or three (15%) small cell clusters separated by an average distance of 50 µm, which only at later stages of epiboly coalesced to form a single cluster (*Figure 8B–D*). Thus, it remained unknown how cells located far apart in the initially dispersed small DFC clusters approached to each other to establish adhesive contacts. To address this question, we followed the movement of DFCs by in vivo microscopy. Our time-lapse image analysis revealed that DFCs located at the edges of distant cell clusters initiated adhesive contacts by sending out long polarised protrusions that resulted in the establishment of long-term adhesion between cells, allowing DFC clustering (*Figure 8E*; *Figure 8—video 1*). The protrusions involved in this process differed from other protrusions formed by DFCs not only by their longer length, but also by their increased lifetime and alignment between clusters (*Figure 8F–I*). Collectively, these results indicate that long-range cell-cell contact interactions mediated by polarised protrusions function as seeds for the establishment of stable adhesion between DFCs. Remarkably, using transgenic embryos expressing a fluorescent biosensor for F-actin only in DFCs, we observed a temporal connection between the formation of protrusion-mediated adhesive contacts and the loss of apical attachments. Subsets of small clusters of DFCs that contained several cells with apical attachments began to detach from extra-embryonic tissues only after initiating protrusion-mediated adhesive contacts with the central cluster of attached DFCs (*Figure 8J* – 0–16 min; *Figure 8—video 2*). Subsequently, they completed the delamination process when fully integrated into the central DFC cluster (*Figure 8J* – 23–47 min; *Figure 8—video 2*). Taken together, these findings indicate that cell-cell contact interactions between DFCs integrate detached cells into the vegetal movement of attached cells to guide the clustering and directional vegetal movement of the entire DFC collective.

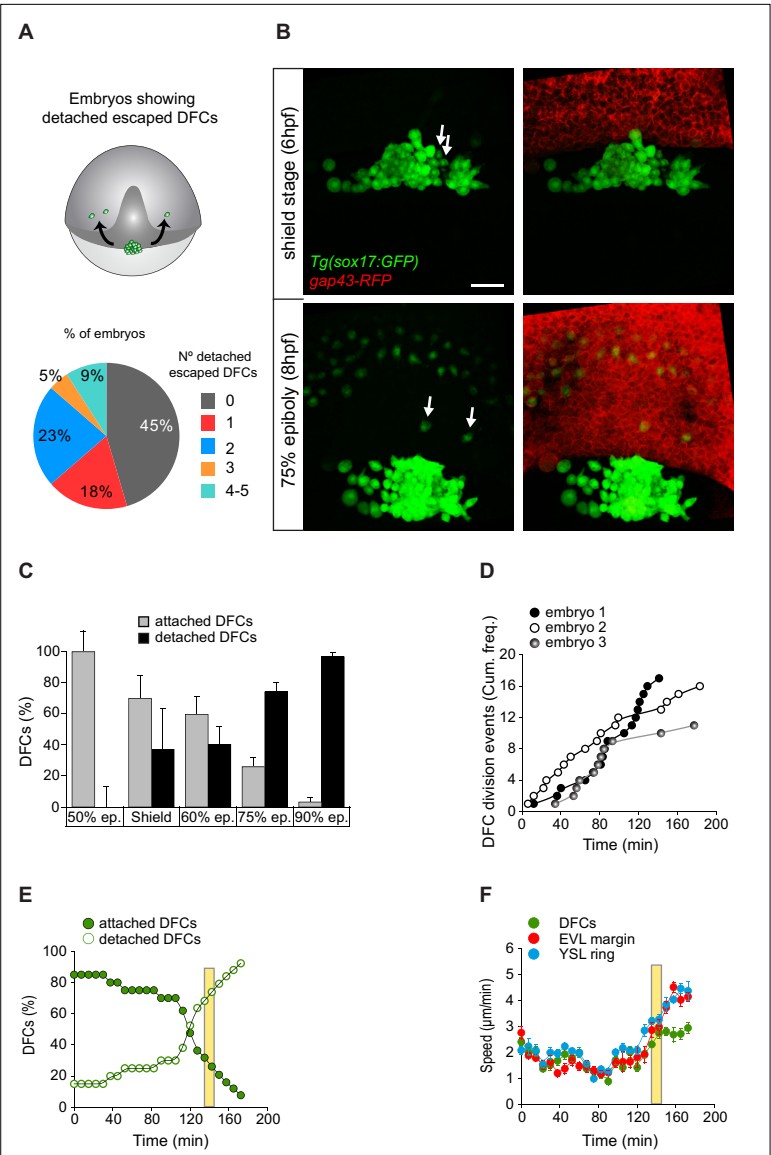

**Figure 6.** Dorsal forerunner cells (DFCs) move to the vegetal pole as a collective despite the increase of detached cells. (**A, B**) Individual detached DFCs can leave the main cluster towards the deep cell layer (DCL) during normal development. (**A**) Schematic diagram showing events of escape (top) and quantification of the percentage of embryos showing escape events of defined numbers of detached DFCs after in vivo imaging of Tg(*sox17::GFP*) embryos (bottom, n = 22 embryos). (**B**) Dorsal views of confocal z-stack maximum projections at shield stage (top) and 75 % epiboly (bottom) of a representative Tg(*sox17::GFP*) embryo injected with *gap43-RFP* mRNA, labelling DFCs in green and the plasma membrane of all cells in red (extracted from *Figure 6—video 1*). Arrows indicate two detached DFCs leaving the main cluster towards the DCL. These cells then mimic the behaviour of endodermal cells and differentiate into endodermal tissue derivatives (*Figure 6—figure supplement 1*). Scale bar, 50 μm. (**C–E**) Origin and progressive increase of detached DFCs during development. (**C**) Quantification of attached and detached DFCs between 50% and 90% epiboly (5.3–9 hpf), as determined from fixed embryos stained with phalloidin and zonula occludens one protein (ZO-1) (see *Figure 1C and D*). Values correspond to means ± s.d. (n = 5–11 embryos per stage). (**D**) Kinetic of cumulative frequency of DFC division events (n = 3 individual representative embryos). (**E**) In vivo kinetics of attached and detached DFCs in a single representative living Tg(*sox17::GFP*) embryo injected with *lifeactin-mcherry* mRNA to label F-actin. Plot shows the percentage of attached (filled green circles) and detached (empty green circles) DFCs over time. (**F**) In vivo progression in the movement speed of the centre of mass of the DFC cluster (green), the enveloping layer (EVL) margin (red), and the actomyosin ring of the yolk syncytial layer (YSL) (blue) (n = 10 DFCs, 5 EVL cell margins, and 4 actomyosin ring points from the same representative embryo as in E). Values correspond to means ± s.d. In E and F, movies started

*Figure 6 continued on next page*

*Figure 6 continued*

at shield stage and extended until 90 % epiboly. The vertical yellow bar indicates the stage when the movement of DFCs uncouples from the vegetal movement of the EVL and YSL (around 80 % epiboly, when detached DFCs reach ~80%). Animal is to the top in all image panels. Source data for all plots are provided in *Figure 6—source data 1*.

The online version of this article includes the following video, source data, and figure supplement(s) for figure 6:

**Source data 1.** Source data for *Figure 6*.

**Figure 6—video 1.** Dorsal forerunner cells (DFCs) can leave the main cluster and internalise into the deep cell layer (DCL) (related to Figure 6A and B).

https://elifesciences.org/articles/66483/figures#fig6video1

**Figure supplement 1.** Dorsal forerunner cells (DFCs) have endodermal potential and acquire endodermal fate after internalising into the deep cell layer (DCL).

## Discussion

Here, we show a previously unexplored mechanism of morphogenesis of a small group of progenitor cells that stems from the mechanistic link between the process of epithelial delamination underlying progenitor specification and the directed movement of adjacent extra-embryonic tissues. This morphogenetic mechanism safeguards progenitors from undesired losses while guiding their directed motion and allocation as a cluster at the site of organ differentiation. In zebrafish, DFCs are the laterality organ progenitors and arise by delamination from the EVL, an extra-embryonic surface epithelium that protects the early embryo and at later stages gives rise to the periderm (*Kimmel et al., 1990*). When DFCs are formed, the EVL epithelium spreads from the equator to the embryo's vegetal pole during the movement of epiboly in conjunction with the YSL, an extra-embryonic syncytium to which the vegetal margin of the EVL is tied. Remarkably, the apical constriction process underlying the delamination of DFC progenitors follows a temporal progression that allows these cells to retain long-term apical attachments with the EVL and YSL as they spread towards the vegetal pole during epiboly. Apical attachments work as tissue connectors that couple DFCs with the vegetal spreading of extra-embryonic tissues, guiding their motion towards the site of differentiation at the vegetal pole. In contrast with previous hypotheses indicating that autonomous motility (*Ablooglu et al., 2010*) and contact-mediated repulsive interactions with the marginal DCL (*Zhang et al., 2016*) drive DFC vegetal movement, we show that mechanical drag by extra-embryonic tissue movement provides the critical guidance cues for DFC directed locomotion. Remarkably, this guidance mechanism is mediated by apical attachments that stem from an incomplete process of delamination, allowing DFCs to coordinate their movement with the adjacent extra-embryonic tissues. Epithelial delamination is a conserved mechanism to generate new mesenchymal cell types in various developmental contexts (*Shook and Keller, 2003*; *Thiery et al., 2009*). Here, we show that besides this canonical function, incomplete delamination serves as a generic mechanism for coordinated tissue movement during development, driving the allocation of newly formed mesenchymal cell groups.

Apical attachments guiding DFC movement arise from a process of apical constriction that is asynchronous among DFCs. We show that such asynchrony generates two distinct populations of progenitors, one holding apical attachments and being pulled by the vegetal movement of extra-embryonic tissues, and a second population of detached delaminated DFCs that follows the vegetal movement of apically attached DFCs through DFC-DFC contact mechanisms. In the context of these evolving two populations, having an asynchronous process of cell delamination potentially increases the chances of maintaining a minimal number of apically attached DFCs able to carry the detached DFC population along their movements, a factor that becomes relevant as epiboly progresses and the ratio of attached/detached DFCs decreases. Furthermore, as apical constriction imposes mechanical stress along the epithelial plane (*Heer and Martin, 2017*; *Martin and Goldstein, 2014*), we can speculate that having asynchrony in a collective process of apical constriction promotes the even dissipation of mechanical stress over time and space, protecting the integrity of the epithelium. Future work combining mechanical perturbations/measurements with physical modelling will have to test these hypotheses directly.

Delaminated DFCs are intrinsically motile and can move towards the DCL, being sequestered by the massive internalisation movements of this embryonic cellular domain during gastrulation. Importantly, when DFCs reach the DCL, either during normal development or after transplantation,

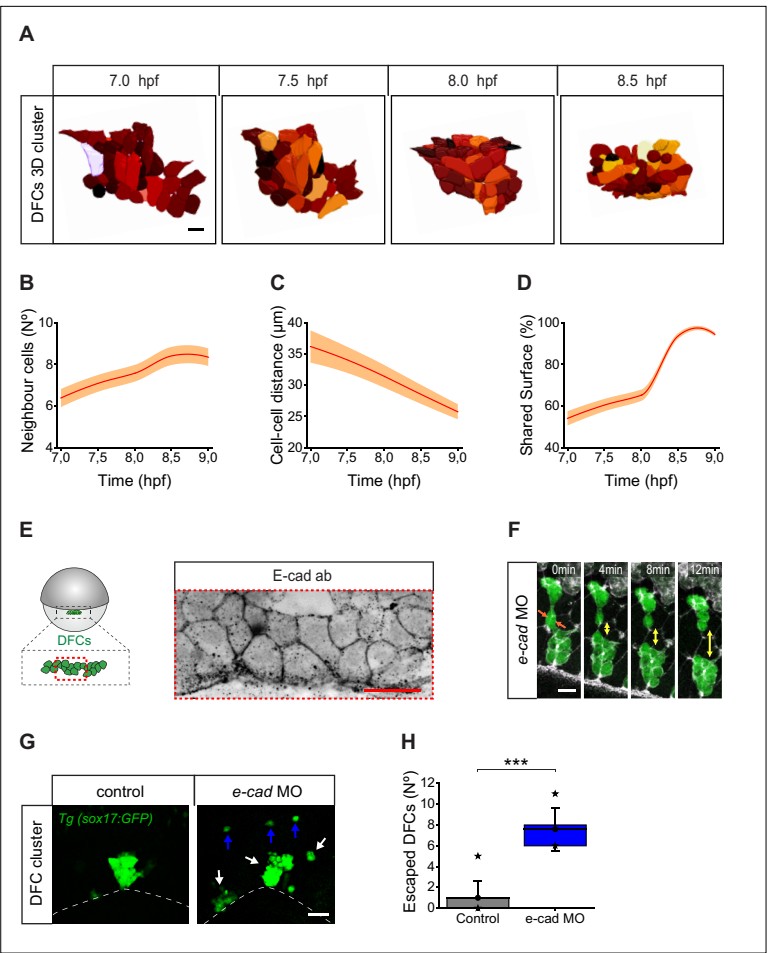

**Figure 7.** Dorsal forerunner cells (DFCs) increase cell-cell contact and cluster compaction during vegetal movement. (**A**) Dorsal view of a 3D cell reconstruction of a single DFC cluster from a *Tg(β-actin::HRAS-EGFP)* embryo at different time points of collective movement. Individual cells are labelled in different colours according to their volume. Scale bar, 20 µm. (**B–D**) Quantification plots of the number of neighbour cells (**B**), and of the distance (**C**) and shared surface (**D**) between neighbouring DFCs (n = 44, 46, 47, 59 cells for 7, 7.5, 8, and 8.5 hpf, respectively, from a single embryo). (**E**) Dorsal view of confocal z-stack maximum projection of a DFC cluster immunostained for E-cadherin (E-cad) in a representative 60 % epiboly embryo. Scale bar, 20 µm. (**F**) Time series of dorsal views of confocal z-stack maximum projections showing the evolution of a small raceme of DFCs in a representative 80 % epiboly Tg(*sox17::GFP*) embryo injected with *e-cad* MO. The double-sided arrows indicate how the DFC raceme becomes apart as DFCs lose their initial adhesive contacts (arrows). Scale bar, 20 µm. (**G**) Dorsal views of confocal z-stack maximum projections showing DFCs from representative 80 % epiboly Tg(*sox17::GFP*) control (left) and *e-cad* MO injected (right) embryos showing the defective collective organisation of DFCs into multiple racemes (white arrows) and the increase of escaped cells (blue arrows). Scale bar, 50 µm. (**H**) Quantification of the number of DFCs escaping from the central cluster towards the animal pole in control and *e-cad* MO injected embryos. The box depicts the interquartile range from 25% to 75% of the data around the average (vertical line inside the box), the whisker depicts s.d., and stars indicate maximum and minimum values (n = 5 embryos for *e-cad* MO and 9 embryos for control conditions). *** (p ≤ 0.001). Animal is to the top in all image panels. Source data for all plots are provided in *Figure 7—source data 1*.

The online version of this article includes the following source data for figure 7:

**Source data 1.** Source data for *Figure 7*.

they follow the endodermal path revealing that DFCs have a previously unrecognised potential to become endoderm that is expressed if they enter the developmental field of the endoderm. Such potential co-option of DFCs by the endodermal DCL reduces the number of progenitors, and this can have a detrimental impact on left-right asymmetry development, increasing the incidence of embryo laterality defects (*Moreno-Ayala et al., 2020*). Here, we show that DFCs transiting the

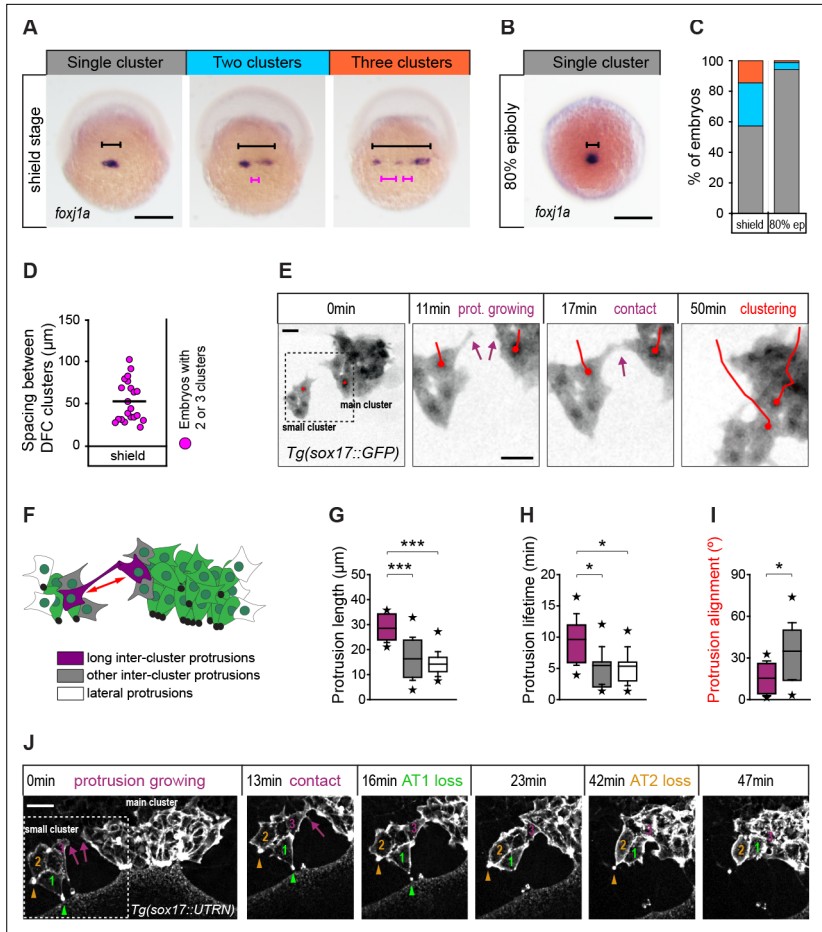

**Figure 8.** Contact interactions between dorsal forerunner cells (DFCs) couple the motion of attached and detached cells promoting a clustered collective movement. (**A**) Spatial distribution patterns of DFCs in dorsal views of wild-type embryos at shield stage revealed by *foxj1a* mRNA expression. During DFC formation, these cells are organised at the dorsal margin as single or multiple clusters spanning different widths (black bracket) along the medio-lateral axis, also showing variable spacing between clusters (purple bracket). (**B**) Same as in (**A**) showing an embryo with a single compact cluster (black bracket) at 80 % epiboly. Scale bars of A and B, 250 μm. (**C**) Quantification of the percentage of embryos showing single (grey), two (light blue), or three (orange) DFC clusters at shield stage and at 80 % of epiboly (n = 110 embryos at shield stage, and 88 embryos at 80 % of epiboly). (**D**) Quantification of the spacing between DFC clusters at shield stage (n = 16 embryos with two or three clusters). (**E**) Temporal confocal series of DFCs from a Tg(*sox17::GFP*) embryo showing long protrusions (purple arrows) contacting DFCs from neighbouring clusters. After the initial contact, DFCs from the lateral small cluster approach and establish adhesive contacts with the main cluster (red lines). Scale bar, 20 μm. (**F**) Schematic diagram showing the different types of protrusions formed by DFCs when two clusters approach and coalesce into a single cluster: long inter-cluster protrusions involved in cluster coalescence (purple), other inter-cluster protrusions (grey), and protrusions formed at lateral sides of the contacting clusters (white). The double-sided red arrow indicates the axis of alignment between approaching clusters. (**G–I**) Quantification of the length (**G**), lifetime (**H**), and alignment (**I**) of the protrusions described in (**F**). Boxes depict the interquartile range from 25% to 75% of the data around the average (vertical line inside the box), whiskers depict s.d., and stars indicate maximum and minimum values (n = 8 long inter-cluster protrusions from three embryos, 15 other inter-cluster protrusions from three embryos, and 14 lateral protrusions from three embryos). *** p ≤ 0.001; * (p ≤ 0.05). (**J**) Temporal confocal series from a representative Tg(*sox17::utrn-GFP*) embryo showing a DFC from a small lateral cluster (cell 3) sending long protrusions and establishing adhesive contacts with the main cluster before other DFCs from the small cluster (1 and 2) lose apical attachments (AT, arrowheads) (extracted from *Figure 8—video 2*). Scale bar, 20 μm. Animal is to the top in all image panels. Source data for all plots are provided in *Figure 8—source data 1*.

The online version of this article includes the following video and source data for figure 8:

**Source data 1.** Source data for *Figure 8*.

*Figure 8 continued on next page*

*Figure 8 continued*

**Figure 8—video 1.** Long polarised protrusions promote the initiation of adhesive contacts between dorsal forerunner cells (DFCs) (related to Figure 8).
https://elifesciences.org/articles/66483/figures#fig8video1

**Figure 8—video 2.** Polarised protrusions promote the establishment of adhesive contacts and coalescence of small dorsal forerunner cell (DFC) clusters before losing apical attachments (related to Figure 8J).
https://elifesciences.org/articles/66483/figures#fig8video2

process of delamination avoid endodermal fate by establishing E-cad-mediated adhesive contacts with the sub-population of attached DFCs before completing the process of delamination. Thus, DFC-DFC contact interactions mediated by E-cad play a dual function, to protect delaminated cells from escaping towards the endoderm and to ensure they move together with the attached DFC population as a collective group. Importantly, DFCs located at long distances establish adhesive contacts by sending long polarised protrusions whose persistence and directionality differ from the other short-lived random protrusions formed by most DFCs. The mechanisms underlying the formation of these directed long protrusions are currently unknown. Notably, recent work shows that migrasomes containing the chemokine ligands Cxcl12a/b become enriched in the extracellular space surrounding DFCs (*Jiang et al., 2019*), thus opening the possibility that long polarised protrusions are a manifestation of a chemoattraction mechanism mediated by these guidance molecules. It is also remarkable the observation that small sub-groups of DFCs delaminate only after establishing adhesive contacts with the main DFC cluster suggesting a mechanistic coupling between cell adhesion, cell motility, and the process of delamination. Both emerging possibilities, the guidance of contact-mediated DFC clustering through environmental chemical cues and the coordination of cellular behaviours at a supra-cellular scale, are interesting aspects to explore in future work. Together, these findings provide a novel developmental function for E-cad, mediating the transfer of movement information between two distinct populations of progenitor cells defined by their state in the delamination process: attached (transiting delamination) and detached (delaminated). Remarkably, motion transfer from attached to detached DFCs resembles the leader-to-follower motion transmission observed in events of collective cell locomotion, many of which also require cadherin-dependent cell-cell contact interactions (*Friedl and Gilmour, 2009*; *Reig et al., 2014*; *Theveneau and Mayor, 2013*). In the case of DFCs, the role of E-cad in motion transmission complements the previously described function in clusterisation (*Hong and Dawid, 2009*; *Matsui et al., 2011*; *Oteiza et al., 2010*), and together ensure that progenitors reach the site of terminal differentiation in a sufficient number and organised as a tight collective, a prerequisite to proceed into further stages of organogenesis (*Oteiza et al., 2010*; *Oteiza et al., 2008*). It is worth mentioning that the proposed mechanism of DFC leader-to-follower locomotion works until the percentage of attached cells falls below the 20 % limit, around 85–90% of epiboly. After this stage, the DFC cluster loses its physical connections with the YSL and EVL and begins an internal cellular reorganisation process leading to rosette formation and organ differentiation (*Oteiza et al., 2008*). During this transition, the DFC cluster continues a vegetal displacement until the end of epiboly but at a slower rate and becomes covered by the DCL. The mechanisms driving this subsequent migratory phase are unknown and could involve contact interactions with the DCL margin.

Developmental cases of cells being dragged through physical bonds with adjacent tissues have recently begun to be reported. Among them, *Caenorhabditis elegans* primordial germ cells internalise during gastrulation due to cohesive contacts that these cells establish with the moving ingressing endoderm (*Chihara and Nance, 2012*). In annual killifish, the embryonic DCL spreads during epiboly as a result of adhesive contacts with the basal epithelial domain of the expanding EVL (*Reig et al., 2017*). In zebrafish, the vegetal spreading of the EVL during epiboly follows the autonomous movement of the YSL to which is tightly bonded at its margin by TJ complexes (*Behrndt et al., 2012*; *Betchaku and Trinkaus, 1978*; *Koppen et al., 2006*; *Schwayer et al., 2019*). Therefore, mechanical drag is an emergent mechanism of cell movement whose extent and impact during embryo morphogenesis needs to be further investigated. Here, we show that drag-mediated locomotion is a crucial driver of organogenesis that emerges at the interface of embryonic and extra-embryonic cellular domains highlighting the essential role of mechanical

information from extra-embryonic tissues in driving early embryo development (*Christodoulou et al., 2019*; *Hiramatsu et al., 2013*; *Reig et al., 2017*).

# Materials and methods

## Key resources table

| Reagent type (species) or resource | Designation | Source or reference | Identifiers | Additional information |
|---|---|---|---|---|
| Strain, strain background (*Danio rerio*, both sex) | AB wild type | ICBM-University of Chile | RRID:ZDB-GENO-960809-7 | |
| Genetic reagent (*Danio rerio*, both sex) | Tg(*sox17::utrn-GFP*) | *Woo et al., 2012* DOI: 10.1083/jcb.201203012. | RRID:ZFIN-ALT-120911-1 | |
| Genetic reagent (*Danio rerio*, both sex) | Tg(*sox17::GFP*) | *Sakaguchi et al., 2006* DOI: 10.1242/dev.02581 | RRID:ZFIN-ALT-061228-2 | |
| Genetic reagent (*Danio rerio*, both sex) | Tg(*β-actin::HRAS-EGFP*) | *Cooper et al., 2005* DOI: 10.1002/dvdy.20252 | | |
| Genetic reagent (*Danio rerio*, both sex) | Tg(*actb1::myl12.1-eGFP*) | *Behrndt et al., 2012* DOI: 10.1126/science.1224143 | RRID:ZFIN-ALT-130108-2 | |
| Genetic reagent (*Danio rerio*, both sex) | Tg(*actb1::mCherry-utrCH*) | *Behrndt et al., 2012* DOI: 10.1126/science.1224143 | | |
| Antibody | (Mouse monoclonal) anti-ZO1 | Thermo Fisher Scientific | Cat# 339100 RRID:AB_2533147 | (1:200) |
| Antibody | (Rabbit polyclonal) anti-Phospho-Myosin Light Chain 2 (Ser19) | Cell Signaling | Cat# 3671 RRID:AB_330248 | (1:200) |
| Antibody | (Rabbit polyclonal) anti-Cdh1 | *Maitre et al., 2012* DOI: 10.1126/science.1225399 | MPI-CBG (#174) | (1:200) |
| Antibody | (Goat polyclonal) anti-mouse Alexa Fluor 488 | Thermo Fisher Scientific | Cat# A-11001 RRID:AB_2534069 | (1:200) |
| Antibody | (Goat polyclonal) anti-rabbit Alexa Fluor 568 | Thermo Fisher Scientific | Cat# A-11011 RRID:AB_143157 | (1:200) |
| Recombinant DNA reagent | pCS2-*Gap43-RFP* (plasmid) | *Reig et al., 2017* DOI: 10.1038/ncomms15431 | | Membrane-bound RFP |
| Recombinant DNA reagent | pCS2-*lifeACT-RFP* (plasmid) | *Behrndt et al., 2012* DOI: 10.1126/science.1224143 | | Actin-RFP |
| Recombinant DNA reagent | pCS2-*GFP-zo1-1b* (plasmid) | *Schwayer et al., 2019* DOI: 10.1016/j.cell.2019.10.006 | | Directed to apical junctions |
| Recombinant DNA reagent | *N-ter(1-300aa)-Mypt1* (plasmid) | *Jayashankar et al., 2013* DOI: 10.1371/journal.pone.0075766 | | Inhibits actomyosin network |
| Recombinant DNA reagent | pCS2-*h2b-GFP* (plasmid) | *Keller et al., 2008* DOI: 10.1126/science.1162493 | | Nuclear GFP |
| Recombinant DNA reagent | pCS2-*foxj1a* (plasmid) | *Neugebauer et al., 2009* DOI: 10.1038/nature07753 | TAAATCGCAGCTCTTCCT TCCAACG | Foxj1a sequence in pCS2 backbone for riboprobe synthesis |
| Sequence-based reagent | Cadherin-1 (cdh1 MO) | This paper | Gene Tools | TAAATCGCAGCTCTTCCTTCCAACG |
| Commercial assay or kit | mMESSAGE mMACHINE SP6 Transcription Kit | Thermo Fisher Scientific | Cat# AM1340 | |
| Software, algorithm | Fiji | *Schindelin et al., 2012* DOI: 10.1038/nmeth.2019 | RRID:SCR_002285 | https://imagej.net/Fiji |
| Software, algorithm | MATLAB | MATLAB Software | RRID:SCR_001622 | https://la.mathworks.com/products/matlab.html |

*Continued on next page*

*Continued*

| Reagent type (species) or resource | Designation | Source or reference | Identifiers | Additional information |
|---|---|---|---|---|
| Software, algorithm | Volocity | Quorum Technologies Inc | RRID:SCR_002668 | https://quorumtechnologies.com/ |
| Software, algorithm | Origin | OriginLab | RRID:SCR_014212 | https://www.originlab.com/ |

## Fish strains and maintenance

Zebrafish (*Danio rerio*) strains were maintained and raised according to previously published procedures (*Westerfield, 2000*). Embryos were grown in E3 solution at 28 °C and staged according to morphology. Fish care and procedures were approved by the Ethical Review Committee and comply with the Animals Scientific Procedures Act 0466 (Protocol CBA#0466 FMUCH). Zebrafish strains used were: wild-type AB, Tg(*actb1::myl12.1-eGFP*) (*Behrndt et al., 2012*), Tg(*sox17::utrn-GFP*) (*Woo et al., 2012*), Tg(*sox17::GFP*) (*Sakaguchi et al., 2006*), Tg(*actb1::mCherry-utrCH*) (*Behrndt et al., 2012*), Tg(*actb1::myl12.1-eGFP; actb1::mCherry-utrCH*), and Tg(*β-actin::HRAS-EGFP*) (*Cooper et al., 2005*).

## Morpholino and mRNA injections

Synthetic mRNA was produced using the SP6 mMessage mMachine kit (Thermo Fisher Scientific). Glass capillaries (BF100-98-15, Sutter Instruments) were pulled using a needle puller (P-97, Sutter Instruments) and mounted on a microinjection system (Picospritzer III, Parker Hannifin). Embryos were microinjected at the one-cell stage as previously described (*Barth and Wilson, 1995*), unless stated otherwise: 100 pg of *Gap43-RFP* (*Reig et al., 2017*) or 40 pg of *lifeACT-RFP* (*Behrndt et al., 2012*) mRNA or *h2b-GFP* (*Keller et al., 2008*) were injected as a counterstain for whole embryo visualisation; 20 pg of *zo1-GFP* mRNA was injected to label apical junctions during DFC delamination (*Schwayer et al., 2019*); 75 pg of *N-ter(1-300aa)-Mypt1* mRNA (*Jayashankar et al., 2013*) was injected into the yolk cell at 3.3 hpf for functional inhibition of the actomyosin network in the YSL; and 2 ng of *cdh1* MO (5'- TAAATCGCAGCTCTTCCTTCCAACG -3', GeneTools) (*Maitre et al., 2012*) was injected to abrogate *cdh1* function.

## In situ hybridisation

Whole mount in situ hybridisation was performed as previously described (*Barth and Wilson, 1995*). Antisense RNA probes were synthesised from partial cDNAs of *foxj1a* (*Neugebauer et al., 2009*). After staining, embryos were embedded in 70 % glycerol and photographed on a Nikon SMZ100 dissecting scope using a Nikon digital camera with NIS software. Images were imported into Adobe Photoshop. For these and all other experiments, image manipulation was limited to adjustment of levels, colour balance, contrast, and brightness.

## Immunohistochemistry

Embryos between 50% and 90% of epiboly (5.3–9 hpf) were fixed and stained as described previously (*Oteiza et al., 2010*). Embryos were mounted on agarose-coated dishes embedded in 1 % low melting point agarose. Samples were imaged on a Leica TCS LSI Confocal microscope with HCS software using a 5 × objective and 488/520 ($\lambda$ exc/ $\lambda$ em) lasers. The following antibodies and dilutions were used: mouse anti-ZO-1 (339,100 Invitrogen, 1:200), anti-pMLC2 (3,671 Cell Signaling, 1:200), anti-Cdh1 (MPI-CBG #174, 1:200), goat anti-mouse Alexa Fluor 488 (A-11001, Thermo Fisher Scientific), and goat anti-rabbit Alexa Fluor 568 (A-110011, Thermo Fisher Scientific).

## Whole embryo confocal imaging

Tg(*sox17::GFP*) embryos injected with 50 pg of *gap43-RFP* mRNA were mounted in 0.5 % low melting point agarose in a custom-designed chamber at either dome or 50 % epiboly stage. The temperature was kept constant at 28 °C throughout the imaging experiment using a temperature control system. Whole embryo in vivo microscopic imaging was performed in a Leica TCS LSI Confocal microscope with HCS software using a 5 × objective and 488/520 ($\lambda$ exc/ $\lambda$ em) lasers.

## Identification of DFCs expressing the sox17::GFP transgene

Although both DFCs and endodermal cells express *sox17::GFP*, DFCs start expressing this transgene slightly earlier than endodermal cells and at higher levels. We took advantage of this feature and identified DFCs at early stages, from around 50 % epiboly. These identified DFCs not only expressed *sox17::GFP*, but also position themselves ahead of the margin of the DCL. Also, at early stages they are attached to the EVL, show a larger size than endodermal cells, and often organise in one, two, or three clusters (see *Figure 8A*). In contrast, endodermal cells that express the *sox17::GFP* transgene are located within the DCL margin (or in anterior positions) and show a dispersed scattered organisation. Importantly, DFCs identified according to these criteria could be followed over time after detachment from the EVL/YSL by time-lapse microscopy to faithfully characterised their behaviour and fate, ruling out any possibility that they could be mistaken for native endodermal cells.

## Isochronic transplantation

Tg(*sox17::GFP*) and Tg(*sox17::GFP*) embryos injected with 50 pg of *lifeactin-mCherry* mRNA at one-cell stage were used as host and donor embryos, respectively. Host and donor embryos were dechorionated manually at 30 % epiboly stage, and cultured in sterile 0.3× Danieau buffer in petri dishes coated with 1 % agarose. During transplantation, embryos were arranged over a 1 % agarose multi-well camera filled with sterile 1× Danieau buffer to increase the concentration of calcium ions and thus allow the proper sealing of the transplant point. DFCs from donor embryos were removed with a 20-µm-diameter glass capillary between shield and 70 % epiboly stage, and then transplanted to the most marginal paraxial region of deep blastoderm of host embryos at equivalent stages. After 15 min, DFC-transplanted host embryos were transferred to a solution of sterile 0.3× Danieau buffer supplemented with penicillin and streptomycin. Later, embryos were fixed in 4 % PFA for 2 hr at room temperature and observed in a Leica TCS LSI Confocal microscope with HCS software using a 5× objective and 488/520 ($\lambda$ exc/ $\lambda$ em) lasers.

## High-resolution confocal imaging

Tg(*sox17::GFP*) and Tg(*sox17::utrn-GFP*) embryos were used for high-resolution confocal imaging of cell protrusions. Embryos were imaged from shield stage onwards in a Volocity ViewVox spinning disc (Perkin Elmer) coupled to a Zeiss Axiovert 200 confocal microscope using a Plan-Apochromat 40× /1.2 W lasers 488/520, 568/600, and 647/697 nm ($\lambda$ exc/ $\lambda$ em). Processing and analysis of digital images were performed using Fiji (*Schindelin et al., 2012*), MATLAB (MATLAB 2014), Volocity (Improvision), and Adobe Photoshop.

## Laser ablation

Mechanical disruption of the actomyosin ring within the YSL was performed as previously described by conducting laser ablation on a UV laser ablation setup (*Behrndt et al., 2012*) equipped with a Zeiss 63 × 1.2 NA water immersion lens using Tg(*actb1::myl12.1-eGFP*) embryos. Embryos were mounted at 50 % epiboly (5.3 hpf) and the YSL actomyosin cortex close to the EVL margin was repeatedly ablated by applying 10 UV pulses at 1000 Hz on a rectangular ROI. Suboptimal ablation intensity was applied to just disrupt the cortical actomyosin flux and marginal ring, and avoid the activation of a wound response within the yolk cell. The kinetic of EVL cells, DFCs, and YSL marginal actomyosin network adjacent to the disrupted cortex was compared with the kinetic of close neighbouring tissues showing intact regions of the actomyosin network as an internal control. Cortical laser ablation of the EVL was performed parallel to the EVL margin and perpendicular to the EVL actomyosin cortex by applying 25 UV pulses at 1000 Hz along a 10 µm line. Retraction of apical ties and DFCs were quantified from maximum *z*-projections images.

## Cell protrusion analysis

Quantification of cell protrusions was assessed by making volumetric binary masks from 4D confocal images using manual threshold to keep the presence of protrusions and avoid artefacts. Each volume was reduced to a 2D cluster mask by z-projection based on maximum intensity. Manual correction allowed to keep protrusion zones that were occluded due to dimensional reduction. The Fiji tool Local Thickness was used to generate a central mask to describe the cluster central zone without the protrusions. Masks containing just the protrusions zones were obtained by subtracting cluster masks and

central masks. Each protrusion was defined with a protrusion axis that connected the base and tip of the protrusion. A long axis connected the cluster centroid and the base of each protrusion. Individual protrusion orientation was described calculating the angle between the long axis and the protrusion axis. For single long protrusion analysis, protrusion were selected by visual inspection in different regions of DFC clusters (*Figure 8F*). Using Fiji software (*Schindelin et al., 2012*), protrusion length was calculated as the maximum distance between the centre of cell nucleus and the tip of protrusion. Protrusion timing was calculated by visual identification as the total time spent among protrusion formation and complete retraction. Protrusion alignment was measured as the angle formed among a vector that passes through the two nuclei's centre of DFC that will generate a contact between two clusters and the vector formed among centre of nucleus a tip of protrusion.

## Cell segmentation

Images obtained from in vivo imaging of *gap43-RFP* or *ZO1-GFP* mRNA injected embryos were segmented manually using Wacom Cintiq Touch Screen tablet (Wacom) and Fiji software (*Schindelin et al., 2012*). A maximum z-projection using Fiji was first applied to each xyz stack to obtain a 2D sequence and simplify the xy segmentation of cell boundaries. To perform segmentation in the yz plane, z-projections containing the central region of cell volume of marginal and submarginal DFCs were selected and reconstructed using Fiji.

## 3D reconstruction of DFCs

Contours of deconvolved z-stacks time points of DFCs obtained from in vivo imaging of Tg(*β-actin::HRAS-EGFP*) were segmented manually using the Wacom Cintiq/Cintiq-15X Touch Screen tablet and Interactive Pen Display (Wacom). Binary masks of segmented cell were generated with a custom-made macro written for the public domain image analysis software Image-SXM (http://www.liv.ac.uk/~sdb/imageSXM). Cellular contours in 3D were improved by the application of a modified version of an active surface model (*Kass et al., 1988*). Briefly, an initial surface mesh was computed from the manually segmented DFCs contours. The active surface model was applied and supervised interactively to set the appropriate combination of elasticity, rigidity, viscosity, external force, and iteration coefficients, to produce a final surface model that assimilates closely the morphology of the cellular contour. Parameters of DFCs interaction inside the cluster configuration were calculated from these surface model.

## Cell tracking

Cell movement was tracked by following the cell's centre in Tg(*sox17::GFP*) embryos, and the nucleus in *H2B* mRNA injected embryos. YSL tracking was performed by following arbitrary landmarks inside the marginal actomyosin network. Tracking of apical attachments was assessed by following actin-rich zones localised in the apical face of DFCs. Tracking was performed manually in 2D from maximum z-projections using Fiji plugin MTrack J. Speed and persistence (ratio of displacement to trajectory length) of movement was then calculated from tracking data.

## Statistical analysis

All experiments were performed at least three times. Unless indicated, plots display the mean and the standard deviation. Statistical inference analysis was conducted initially by a *Shapiro-Wilk* test to assess if the data were normally distributed. Two-sample *F*-test of equality of variance was applied to assess if the samples had the same variance. Significance for two groups with normal distribution was calculated through a two-tail *t*-test and the p-value was selected depending if the dataset had or not equality of variance. For other distributions, non-parametric *Kolmogorov-Smirnov* or Mann-Whitney tests were applied. All statistical analyses were conducted using the Origin 2016 (OriginLab). ns: p > 0.05, *: p ≤ 0.05, **: p ≤ 0.01, and ***: p ≤ 0.001.

## Acknowledgements

We thank the bioimaging and zebrafish facilities of ICBM-U.Chile and IST Austria for continuous support. We also thank Felipe Santibañez and Mauricio Cerda for providing algorithms for image analysis.

## Additional information

### Funding

| Funder | Grant reference number | Author |
|---|---|---|
| Fondo Nacional de Desarrollo Científico y Tecnológico | 1190806 | Eduardo Pulgar<br>Rodrigo Soto<br>Steffen Härtel<br>Miguel L Concha |
| Fondo Nacional de Desarrollo Científico y Tecnológico | 1161274 1181823 | Steffen Härtel<br>Miguel L Concha |
| Instituto Milenio de Neurociencia Biomedica | ICN09_015 | Steffen Härtel<br>Miguel L Concha |
| Millennium Nucleus Physics of Active Matter from ANID | ANID | Susana Márquez<br>Rodrigo Soto<br>Miguel L Concha |
| Fondo de Equipamiento Cientifico y Tecnologico | EQM130051 | Miguel L Concha |
| Fondo de Financiamiento de Centros de Investigacion en Areas Prioritarias | 15150012 | Miguel L Concha |
| Fondo Nacional de Desarrollo Científico y Tecnológico | 3160478 | Eduardo Pulgar |
| Comisión Nacional de Investigación Científica y Tecnológica | PIA ACT-1402 | Rodrigo Soto<br>Steffen Härtel<br>Miguel L Concha |
| Comisión Nacional de Investigación Científica y Tecnológica | PIA ACT192015 | Miguel L Concha |
| Comisión Nacional de Investigación Científica y Tecnológica | REDES170212 REDES130020 | Eduardo Pulgar<br>Steffen Härtel<br>Miguel L Concha |
| H2020 European Research Council | Advanced grant 742573 | Carl-Philipp Heisenberg |

The funders had no role in study design, data collection and interpretation, or the decision to submit the work for publication.

### Author contributions

Eduardo Pulgar, Conceptualization, Data curation, Formal analysis, Funding acquisition, Investigation, Methodology, Project administration, Resources, Supervision, Validation, Visualization, Writing – original draft, Writing – review and editing; Cornelia Schwayer, Investigation, Methodology, Writing – review and editing; Néstor Guerrero, Investigation, Methodology; Loreto López, Investigation; Susana Márquez, Methodology, Software; Steffen Härtel, Funding acquisition, Methodology, Supervision; Rodrigo Soto, Methodology, Software, Writing – review and editing; Carl-Philipp Heisenberg, Conceptualization, Funding acquisition, Methodology, Resources, Supervision, Writing – review and editing; Miguel L Concha, Conceptualization, Funding acquisition, Investigation, Methodology, Project administration, Resources, Supervision, Visualization, Writing – original draft, Writing – review and editing

### Author ORCIDs

Carl-Philipp Heisenberg http://orcid.org/0000-0002-0912-4566
Miguel L Concha http://orcid.org/0000-0003-3353-9398

## Ethics

Fish care and procedures were approved by the Ethical Review Committee and comply with the Animals Scientific Procedures Act 0466 (Protocol CBA#0466 FMUCH).

## Decision letter and Author response

Decision letter https://doi.org/10.7554/eLife.66483.sa1
Author response https://doi.org/10.7554/eLife.66483.sa2

---

# Additional files

## Supplementary files

• Transparent reporting form

## Data availability

All data generated or analysed during this study are included in the manuscript and supporting files. Source data files have been provided for Figures 1, 2, 3, 4, 5, 6, 7 and 8.

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
