## [Decision Letter]

**Acceptance summary:**

We found your work of general interest with an important contribution for the fields of cell biology and morphogenesis. Your work highlights a novel mechanism of cell movement where cells pull a second group of cells leading into a coordinated movement. We congratulate you for this nice piece of work.

**Decision letter after peer review:**

Thank you for submitting your article "Apical contacts stemming from incomplete delamination guide progenitor cell allocation through a dragging mechanism" for consideration by *eLife*. Your article has been reviewed by 3 peer reviewers, one of whom is a member of our Board of Reviewing Editors, and the evaluation has been overseen by Richard White as the Senior Editor. The reviewers have opted to remain anonymous.

Essential revisions:

(1) The authors describe how detached DFCs reattach to the main cluster via polarized protrusions (Figure 6). The existence of preferentially polarized protrusions even before cell contact suggests that there are long range cues that help direct cell processes, but no attempt has been made to identify these cues. Even if the authors do not perform extra experiments, could they discuss possible players in their discussion?

The findings of ECad are not that surprising because ECad is already known to be involved in DFC migration. It seems from Figure 6 that ECad MO doesn't completely stop clustering, suggesting other adhesive proteins are involved and this is not explored further. It would be great to provide some examples and possibly test the implication of these factors.

(2) It is not clear how the different mechanisms of escaped DFCs finding their way back to joining the main cluster fit together. How many escaped DFCs escape as single cells? How often does 6F occur? Are small clusters found in 6D formed via individual escaped DFCs joining together, or by an event like 6F except delamination happened before the cluster attached itself to the main cluster?

(3) Figure 1C and D aims to show that ZO1 is accumulated at the junctions between DFC and EVL cells. In order to reach that conclusion they need to show images in which the DFC are easily identifiable (e.g. Tg(Sox17::GFP)) in addition of the ZO1 expression. Unfortunately in Figure 1C they merge the ZO1 and phalloidin channels and in Figure 1D they do not show the Tg(Sox17::GFP).

(4) The conclusion that apical contraction is mediated by actomyosin dynamic is the weakest aspect of the manuscript, and at the same time the least interesting. First, it is expected that apical contraction is driven by actomyosin dynamic; so, no novelty there. Second, the data here are just corelations and there is no functional experiment that really demonstrates that actomyosin activity is required for apical contraction. Third, the images shown in Figure 3G and corresponding Video 3 are confusing. It is not clear what region of the cell to look at, as the levels of myosin are quite variable across different regions. In addition, Figure 3H shows that the apical area reach almost cero at 7 min, but I cannot see that in the images or video. In conclusion, without stronger evidence we suggest to remove this part, as it does not contribute to the main mechanism explored in this manuscript.

(5) The authors use the data from Figure 3 and Figure S2 to propose that DFC and EVL movements are linked. The problem with this analysis is that the comparison in the movement of DFC and EVL is done at the cell population level. A better analysis could be achieved by analysing single cells; for example the tracks of single DFCc could be compared with the track of neighbour EVL cells. This could show that the similarities between DFC and EVL cells movement decreases with the distance between these two cell populations, if they are really moving together.

(6) The observation that detached cells can cluster with the DFC cells via protrusions and E-cadherin is an interesting observation. However further analysis should be performed. Are the detached cells producing protrusions in random orientations and only those growing in the direction of the DFC cluster become larger or more stable? What about the protrusions produced by the DFC cluster, are longer or more stable if directed to detached DFC cells? How does the protrusions dynamics change with the distance between the detached DFC cells and the DFC cluster? Answering these questions could give an idea about the mechanism by which single cells are reintegrated to the DFC cluster.

(7) The authors conclude that apical constriction of EVL cells depends on actomyosin dynamics. The authors show a compelling correlation between apical constriction and actomyosin accumulation (Figure 2). However, no functional perturbations of actomyosin activity were shown to disrupt apical constriction. Thus, it remains unknown whether apical constriction requires actomyosin activity.

(8) The evidence presented supports, but does not fully demonstrate, the proposed drag mechanism. Manipulation of the YSL actomyosin ring (by myosin dephosphorylation or laser ablation) shows that both DFC and EVL cells require YSL actomyosin ring contraction for vegetal movement. In addition, DFC and EVL cell velocity and directionality indicates a correlation between the movements of the two cell types. Based on these results, it is proposed that apical attachments between the layers allow the EVL to drag the DFCs. To directly test this hypothesis, the authors turn to laser ablations. If the drag hypothesis is correct, I would predict that severing all (or several) of the attachments would result in the EVL leaving the DFCs behind. This would convincingly demonstrate that EVL cells drag DFCs via attachments. However-potentially due to technical reasons-the authors sever only one of the attachments. The DFCs recoil, but the attachment is almost immediately reconnected, and there is no apparent effect on movement of EVL or DFCs. Therefore, it is unclear whether apical attachments are necessary for DFC movement.

(9) The data presented clearly show attachments between DFCs and EVL. However, it also appears that leading DFCs are attached to the YSL actomyosin ring. Experimental perturbations indicate that contractility of the actomyosin ring is required for DFC movement, raising the possibility that the YSL actomyosin ring may 'pull' the DFCs vegetally. The authors should comment on this alternative mechanism.

10) Previous work has also suggested that DFCs have intrinsic motility and respond to chemoattraction (Zhang, et al., 2016. Development; Ablooglu, et al., 2010. Development; Jiang, et al., 2019. Nat Cell Biol). Although the authors characterize protrusive activity of DFCs, no functional evidence is presented to rule out this mechanism. It is also possible that the proposed mechanisms of DFC movement are not mutually exclusive.

(11) It should be clarified that DFCs detach from the EVL ~8-9 hpf, which is part way through their journey to the site of differentiation and organ formation in the tailbud between 10-11 hpf. It may be useful for the authors to speculate on mechanisms that control the later part of this journey, which may provide foundations for future work.

(12) Imaging of apical constriction of EVL cells (Figure 1) to form DFCs is convincing. However, there may be some confusion between epithelial EVL cells and mesenchymal DFCs. It should be clarified that EVL cells are polarized epithelial cells, otherwise it may be unclear to the reader how apical surfaces are defined in these cells. In addition, "DFC apical constriction" (as written in the text) may be confusing since DFCs are considered to be mesenchymal-like cells, likely with front-rear polarity rather than apical-basal polarity.

(13) Since the presented results do not fully 'demonstrate' the proposed drag mechanism, I suggest the authors soften their conclusions.

(14) It is not clear whether "escaper DFCs" (Figure 5) are DFCs or endoderm cells, since the sox17:GFP transgene is expressed in both cell types. The authors should comment on this in the text.

---

## [Author Response]

Essential revisions:1) The authors describe how detached DFCs reattach to the main cluster via polarized protrusions (Figure 6). The existence of preferentially polarized protrusions even before cell contact suggests that there are long range cues that help direct cell processes, but no attempt has been made to identify these cues. Even if the authors do not perform extra experiments, could they discuss possible players in their discussion?

As we have shown in the new quantitative analysis of protrusion formation in Figure 8, DFCs produce a type of protrusion that, compared to other protrusions formed by the same cells, are longer, show an extended lifetime, and are aligned between clusters. These findings suggest that this type of polarised protrusions could be a manifestation of some type of short- or long-range attraction mechanism. Although addressing the molecular nature of this mechanism is beyond the scope of this article, we provide in the discussion a molecular candidate to be addressed in future work based on a recent publication from the Meng and Yu laboratories (Jian et al., 2009; Nature Cell Biology 21: 966-977). This publication shows that migrasomes containing the chemokine ligands Cxcl12a/b are enriched in the extracellular space surrounding DFCs, making it possible that chemokine signalling can function as a guidance mechanism for directing polarised protrusions and clustering of DFCs.

The findings of ECad are not that surprising because ECad is already known to be involved in DFC migration. It seems from Figure 6 that ECad MO doesn't completely stop clustering, suggesting other adhesive proteins are involved and this is not explored further. It would be great to provide some examples and possibly test the implication of these factors.

Indeed, previous work from several laboratories have shown that E-cad is required for DFC cluster formation (Hong and Dawid, 2009 PNAS 106:2230-5; Matsui et al., 2011 PNAS 108:9881-6; Oteiza et al., 2010 Development 137:3459-68). These publications and the present work have also shown that cluster formation is not completely abolished after E-cad knockdown. This finding could be in part related to the fact that E-cad MO experiments (and other published experimental conditions) might only induce a partial loss of E-cad function. But it is also possible the existence of compensatory mechanisms (e.g. increase in the expression of other cadherins) or the involvement of other adhesion molecules. To date there is no description of other adhesion proteins participating in DFC clustering. This is an interesting aspect of DFC clusterisation that has not been the focus of our work and should definitively be addressed in the future. A possible avenue of future work is exploring the role of N-cadherin as preliminary data from our lab shows that N-cad is also expressed in DFCs and that knockdown of this adhesion molecule induces the formation of a smaller, misshapen Kupffer's vesicle and, consequent, a left-right asymmetry defect (data not shown).

(2) It is not clear how the different mechanisms of escaped DFCs finding their way back to joining the main cluster fit together. How many escaped DFCs escape as single cells?

All detached DFCs that escape into the DCL do so as single cells. However, the number of escaped cells varies from embryo to embryo. As shown in the new quantification panel of Figure 6A, only 55% of embryos show escaped DFCs. Among them, the most frequent cases are embryos with 1 (18%) or 2 (23%) escaped cells, whereas embryos with 3 (5%) and 4-5 (9%) escaped cells are much less frequent. In the source data section, we have included additional information related to escaped cells, such as the developmental stage of DFC escaped cells and whether or not a cell escape occurs in association with a cell division event.

How often does 6F occur?

The original Figure 6F (now Figure 8J) showed that a small cluster of DFCs integrated into the central cluster of DFCs before losing their apical attachments to the YSL/EVL. Thus, this figure illustrates a case in which detached DFCs avoid escape by first establishing adhesive contacts with attached DFCs within the small cluster, and then by engaging into the vegetal movement of the central cluster. We provide a new quantification of the frequency of this behaviour in Figure 8. We observed that in the early stages of vegetal movement (shield phase), the population of DFCs was organised as a single cluster of cells in only 57% of embryos. In the remaining 43% of embryos, DFCs initially formed 2 (28%; similar to the configuration shown in Figure 8J) or 3 (15%) small clusters of cells separated by an average distance of 50µm. As shown in Figure 8J (also in Figures 8B-D), the configuration of 2-3 clusters coalesced to form a single cluster only at later stages of epiboly, using the DFC-DFC contact mechanism proposed in the paper (Figures 8B-D,J).

Are small clusters found in 6D formed via individual escaped DFCs joining together, or by an event like 6F except delamination happened before the cluster attached itself to the main cluster?

The small cluster seen in original Figure 6D (new Figure 8E) forms similarly to the cluster shown in Figure 8J. In both cases, the small clusters are the result of an early DFC multi-cluster configuration seen in about 43% of embryos, which coalesce into a single cluster only in the late stages of epiboly. The only difference between the small cluster shown in Figure 8E and 8J is the GFP labelling: in 8E the DFCs are labelled with cytoplasmic GFP whereas the GFP in 8J labels the F-actin, which reveals the DFC apical attachments with the EVL and YSL, which are invisible in 8E.

(3) Figure 1C and D aims to show that ZO1 is accumulated at the junctions between DFC and EVL cells. In order to reach that conclusion they need to show images in which the DFC are easily identifiable (e.g. Tg(Sox17::GFP)) in addition of the ZO1 expression. Unfortunately in Figure 1C they merge the ZO1 and phalloidin channels and in Figure 1D they do not show the Tg(Sox17::GFP).

We have modified Figure 1C to show the staining of ZO1, phalloidin and the *sox17::GFP* transgene in separate channels with different colours, revealing more clearly the specific accumulation of ZO1 (and F-actin) at the apical junctions of the marginal and sub-marginal DFCs with the EVL and YSL. Also, we have incorporated a supplemental figure (Figure 1—figure supplement *1*) where we show a dorsal view of the Tg(*sox17::GFP*) embryo used to take the images shown in panels 1 and 2 of Figure 1C, revealing the punctuated coaccumulation of ZO-1 and F-actin at the DFC-EVL interfaces.

(4) The conclusion that apical contraction is mediated by actomyosin dynamic is the weakest aspect of the manuscript, and at the same time the least interesting. First, it is expected that apical contraction is driven by actomyosin dynamic; so, no novelty there. Second, the data here are just corelations and there is no functional experiment that really demonstrates that actomyosin activity is required for apical contraction. Third, the images shown in Figure 3G and corresponding Video 3 are confusing. It is not clear what region of the cell to look at, as the levels of myosin are quite variable across different regions. In addition, Figure 3H shows that the apical area reach almost cero at 7 min, but I cannot see that in the images or video. In conclusion, without stronger evidence we suggest to remove this part, as it does not contribute to the main mechanism explored in this manuscript.

We agree with the reviewers that the description of actomyosin dynamics is not central to the story and does not contribute to the main mechanism of DFC locomotion. Therefore, we have followed the suggestion and removed this part of the manuscript.

(5) The authors use the data from Figure 3 and Figure S2 to propose that DFC and EVL movements are linked. The problem with this analysis is that the comparison in the movement of DFC and EVL is done at the cell population level. A better analysis could be achieved by analysing single cells; for example the tracks of single DFCc could be compared with the track of neighbour EVL cells. This could show that the similarities between DFC and EVL cells movement decreases with the distance between these two cell populations, if they are really moving together.

The analysis in the original Figure S2 (now part of panels D-G in Figure 3) is in fact a population-level analysis based on single-cell tracking comparing the speed and directionality of DFC movement with the speed and directionality of EVL movement. We have complemented this analysis with a correlation analysis between the movement of the DFC cluster and the movement of the EVL margin (New Figure 3G). Nevertheless, the aim of this analysis was to give a first approximation of the spatio-temporal connection between the movement of DFCs and EVL and we did not intend with this analysis to demonstrate a causal connection between the two processes. To address this point more directly, we have performed a new correlation analysis at the individual cell level between the movement of DFCs and the movement of their apical attachments, which is shown in the new Figure 5E-I. This analysis revealed that the movement of DFCs and their apical attachments are indeed closely correlated, and that this link is lost when the attachments are lost supporting a requirement of this attachments for the directed vegetal movement of DFCs.

(6) The observation that detached cells can cluster with the DFC cells via protrusions and E-cadherin is an interesting observation. However further analysis should be performed. Are the detached cells producing protrusions in random orientations and only those growing in the direction of the DFC cluster become larger or more stable? What about the protrusions produced by the DFC cluster, are longer or more stable if directed to detached DFC cells? How does the protrusions dynamics change with the distance between the detached DFC cells and the DFC cluster? Answering these questions could give an idea about the mechanism by which single cells are reintegrated to the DFC cluster.

We agree with the reviewers that a more in-depth analysis was needed and therefore have performed a quantitative analysis of the dynamics of cell protrusions shown in the new Figure 8 (panels E-I). In the analysis we make a clear distinction between the behaviour of cell protrusions involved in DFC clustering versus the behaviour of protrusions not involved in clustering, which are produced by DFCs both between clusters and on the lateral sides of them. We show that protrusions involved in clustering are longer, show a longer lifetime and also align between clusters, in contrast to the small, short-lived and randomly-oriented nature of the other protrusions. We have incorporated these results into the main text, Figure 8E-I, and mention in the discussion that long polarised protrusions involved in clustering could be a manifestation of a chemo attractive mechanism.

(7) The authors conclude that apical constriction of EVL cells depends on actomyosin dynamics. The authors show a compelling correlation between apical constriction and actomyosin accumulation (Figure 2). However, no functional perturbations of actomyosin activity were shown to disrupt apical constriction. Thus, it remains unknown whether apical constriction requires actomyosin activity.

As explained in point (4), actomyosin dynamics do not contribute to the main mechanism of DFC locomotion, so we have followed the reviewers' suggestion and removed this part of the manuscript.

(8) The evidence presented supports, but does not fully demonstrate, the proposed drag mechanism. Manipulation of the YSL actomyosin ring (by myosin dephosphorylation or laser ablation) shows that both DFC and EVL cells require YSL actomyosin ring contraction for vegetal movement. In addition, DFC and EVL cell velocity and directionality indicates a correlation between the movements of the two cell types. Based on these results, it is proposed that apical attachments between the layers allow the EVL to drag the DFCs. To directly test this hypothesis, the authors turn to laser ablations. If the drag hypothesis is correct, I would predict that severing all (or several) of the attachments would result in the EVL leaving the DFCs behind. This would convincingly demonstrate that EVL cells drag DFCs via attachments. However-potentially due to technical reasons-the authors sever only one of the attachments. The DFCs recoil, but the attachment is almost immediately reconnected, and there is no apparent effect on movement of EVL or DFCs. Therefore, it is unclear whether apical attachments are necessary for DFC movement.

We agree that severing the apical attachments of DFCs would indeed be the ultimate experiment to directly test the requirement of apical attachments for the vegetal movement of DFCs, but, as the reviewers rightly presume, this experiment is not technically possible (laser ablation of the apical attachment inevitably leads to the death of the DFC). However, we argue that the EVL ablation experiments we provide in the article, combined with the “natural” experiments of single isolated DFCs (not mentioned by the reviewers), for which we provide additional analysis to demonstrate the link between the movement of attached DFCs and their apical attachments (Figure 5F-I), strongly support the drag-mediated mechanism. Below we explain in more detail the rationale for these experiments and discuss their results (also included in new version of text).

(i) EVL ablation experiment (Figure 4C-E). In this experiment we transiently released a subset of DFC apical attachments from the influence of EVL spreading by performing laser ablation of a cortical EVL junction located immediately vegetal to DFC apical attachments. We reasoned that if the EVL carries the DFCs through the apical attachments, they should be under tension and, therefore, cutting an EVL junction in a vegetal position to the attachments should result in a transient recoil towards the animal pole of both the apical attachment and body of the DFC, which is what we observed. We also observed that wound repair after ablation caused an accumulation of actin in the EVL that resulted in a pull of the apical attachments towards vegetal and a resulting movement of the DFC in the same direction, corroborating that the apical attachments are capable of transmitting movement information from the EVL to the DFCs to guide their movement.

(ii) “Natural” experiment (Figure 5). We took advantage of the particular spatial configuration that DFCs adopt in some wild-type embryos, in which subsets of cells developed as individuals isolated from the core group of DFCs, to test the requirement of apical attachments for DFC directed vegetal movements. In these “natural” experiments, we observed that isolated individual DFCs devoid of apical attachments moved without persistence and directionality, even towards the animal pole in the opposite direction to the vegetal movement of the YSL/EVL. In contrast, isolated individual DFCs connected with the extra-embryonic tissues by apical attachments showed a persistent and directed movement toward the vegetal pole, mimicking the movement of extra-embryonic tissues. Importantly the persistence and speed of movement of DFCs correlated with the persistence and speed of movement of apical attachments, and this close spatial and temporal relationship between apical attachments and DFC movements was only lost upon detachment (Figure 5F-I), indicating the requirement of physical attachment to extra-embryonic tissues for the persistent directed vegetal movement of DFCs.

Taken together, these findings demonstrate that apical attachments are required to transmit vegetal spreading from extra-embryonic tissues to attached DFCs, guiding their vegetal movement during epiboly. This mechanism appears critical for the guidance of DFC locomotion but we do not claim is the only mechanism, as cell protrusions and contact interactions with the DCL margin may facilitate the guidance and act in coordination with EVL/YSL drag.

9) The data presented clearly show attachments between DFCs and EVL. However, it also appears that leading DFCs are attached to the YSL actomyosin ring. Experimental perturbations indicate that contractility of the actomyosin ring is required for DFC movement, raising the possibility that the YSL actomyosin ring may 'pull' the DFCs vegetally. The authors should comment on this alternative mechanism.

The combined role of EVL and YSL (to which the marginal DFCs are attached) is indeed a central aspect of our proposal. We mention it in several places but, as is evident from the reviewers' comments, it is not sufficiently clear, so we have made changes to the main text and discussion to make this point even clearer.

10) Previous work has also suggested that DFCs have intrinsic motility and respond to chemoattraction (Zhang, et al., 2016. Development; Ablooglu, et al., 2010. Development; Jiang, et al., 2019. Nat Cell Biol). Although the authors characterize protrusive activity of DFCs, no functional evidence is presented to rule out this mechanism. It is also possible that the proposed mechanisms of DFC movement are not mutually exclusive.

We have performed additional experiments to complement the analysis of protrusion formation and to address more directly the role of polarised cell protrusions in DFC vegetal movement, which is now shown in new Figure 2 (panels F-L). To test the requirement of polarised cell protrusions for DFC vegetal movement, we inhibited the formation of actin-based cell protrusions by expressing a dominant negative form of Rac (Rac1-T17N). We showed that DFCs expressing Rac1-T17N did not form lamellar or filopodial protrusions as in controls, but displayed extensive membrane blebbing without directional bias. However, they still retained a persistent collective movement towards the vegetal pole. This result, together with the lack of vegetally-oriented polarised protrusions in the WT, indicates that the primary mechanism of DFC vegetal movement does not require polarised cell protrusions, and that other mechanisms must play a role, in this case, the EVL/YSL drag mechanisms proposed in the paper.

11) It should be clarified that DFCs detach from the EVL ~8-9 hpf, which is part way through their journey to the site of differentiation and organ formation in the tailbud between 10-11 hpf. It may be useful for the authors to speculate on mechanisms that control the later part of this journey, which may provide foundations for future work.

We have clarified this point in the results and also provide some speculation in the discussion, as suggested.

12) Imaging of apical constriction of EVL cells (Figure 1) to form DFCs is convincing. However, there may be some confusion between epithelial EVL cells and mesenchymal DFCs. It should be clarified that EVL cells are polarized epithelial cells, otherwise it may be unclear to the reader how apical surfaces are defined in these cells. In addition, "DFC apical constriction" (as written in the text) may be confusing since DFCs are considered to be mesenchymal-like cells, likely with front-rear polarity rather than apical-basal polarity.

We understand this concern and to avoid confusion we have clarified when possible in the text the epithelial nature of the EVL cells, the mesenchymal nature of the DFCs, and the dual behaviour of EVL cell transiting delamination (DFC progenitors).

13) Since the presented results do not fully 'demonstrate' the proposed drag mechanism, I suggest the authors soften their conclusions.

See discussion in point 8.

14) It is not clear whether "escaper DFCs" (Figure 5) are DFCs or endoderm cells, since the sox17:GFP transgene is expressed in both cell types. The authors should comment on this in the text.

Although both DFCs and endodermal cells express *sox17::GFP*, DFCs start expressing this transgene slightly earlier than endodermal cells and at higher levels. We took advantage of this feature and identified DFCs at early stages, from around 50% epiboly. These identified DFCs not only expressed *sox17::GFP* but also position themselves ahead of the margin of the deep cell layer (DCL). Also, at early stages they are attached to the EVL, show a larger size than endodermal cells, and often organise in 1, 2 or 3 clusters (see Figure 8A). In contrast, endodermal cells that express the *sox17::GFP* transgene are located within the DCL margin (or in anterior positions) and show a dispersed scattered organisation. Importantly, DFCs identified at early stages according to these criteria could be followed over time after detachment from the EVL/YSL by time-lapse microscopy to faithfully characterised their behaviour and fate, ruling out any possibility that they could be mistaken for native endodermal cells. We have incorporated this information into the Material and Methods section.